# A second-order-like optimizer with adaptive gradient scaling for deep learning

**Jérôme Bolte**                                                    *jerome.bolte@ut-capitole.fr*
*Toulouse School of Economics,*
*University of Toulouse Capitole,*
*Toulouse, France*

**Ryan Boustany**                                                  *ryan.boustany@ut-capitole.fr*
*Toulouse School of Economics,*
*University of Toulouse Capitole,*
*Thales LAS France*

**Edouard Pauwels**                                            *edouard.pauwels@ut-capitole.fr*
*Toulouse School of Economics,*
*University of Toulouse Capitole,*
*Toulouse, France*

**Andrei Purica**                                              *andrei.purica@thalesgroup.com*
*Thales LAS France*

**Reviewed on OpenReview:** *https://openreview.net/forum?id=3khtiJDXQW*

## Abstract

In this empirical article, we introduce INNAprop, an optimization algorithm that combines the INNA method with the RMSprop adaptive gradient scaling. It leverages second order information and rescaling while keeping the memory and compute requirements of standard DL methods as AdamW or SGD. INNAprop is evaluated on CIFAR-10, Food101, and ImageNet with ResNets, VGG, DenseNet, and ViT. We also train GPT-2 (OpenWebText) from scratch and with LoRA fine-tuning (E2E). INNAprop consistently offers close performance to AdamW, while performing significantly better in our LLM training experiments, achieving faster convergence and higher accuracy with minimal hyperparameter tuning, even at large scale. Our code is public [1].

## 1 Introduction

As deep learning models grow in size, massive computational resources are needed for training, representing significant challenges in terms of financial costs, energy consumption, and processing time (Susnjak et al., 2024; Varoquaux et al., 2024). According to the UN's Environment Programme Training, the Big Tech sector produced between two and three percent of the world's carbon emissions in 2021; some estimations for the year 2023 go beyond 4%, see the latest Stand.earth reports, and also (Schwartz et al., 2020; Strubell et al., 2020; Patterson et al., 2021) for related issues. For instance, training GPT-3 is estimated to require 1,287 megawatt-hours (MWh) of electricity, which is equivalent to the annual usage of over 100 U.S. households (Anthony et al., 2020; Patterson et al., 2021). OpenAI claimed that the training cost for GPT-4 (Achiam et al., 2023) exceeded 100 million dollars. The PaLM model developed by Google AI was trained for two months using 6144 TPUs for 10 million dollars (Chowdhery et al., 2023). This calls for faster and more cost-efficient optimization algorithms. It also suggests that early stopping (Prechelt, 2002; Bai et al., 2021) in the training phase is a desirable feature whenever possible.

---

[1] https://github.com/innaprop/innaprop

We focus in this work on the computational efficiency during the training phase and consider the problem of unconstrained minimization of a loss function $\mathcal{J} \colon \mathbb{R}^p \to \mathbb{R}$, as follows

$$\min_{\theta \in \mathbb{R}^p} \mathcal{J}(\theta). \tag{1}$$

**Continuous dynamical systems as optimization models.** To achieve higher efficiency, a useful way is to view algorithms as discrete versions of continuous dynamical systems (Ljung, 1977), further developed in (Harold et al., 1997; Benaïm, 2006; Borkar & Borkar, 2008; Attouch et al., 2016; Aujol et al., 2019). In deep learning, this approach is also quite fruitful; it has, in particular, been used to provide convergence proofs or further geometric insights (Davis et al., 2020; Bolte & Pauwels, 2020; Barakat & Bianchi, 2021; Chen et al., 2023a).

In the spirit of Castera et al. (2021), we consider the following continuous-time dynamical system introduced in Alvarez et al. (2002) and referred to as DIN (standing for "dynamical inertial Newton"):

$$\underbrace{\ddot{\theta}(t)}_{\text{Inertial term}} + \underbrace{\alpha\,\dot{\theta}(t)}_{\text{Friction term}} + \underbrace{\beta\,\nabla^2\mathcal{J}(\theta(t))\dot{\theta}(t)}_{\text{Newtonian effects}} + \underbrace{\nabla\mathcal{J}(\theta(t))}_{\text{Gravity effect}} = 0 \tag{DIN}$$

where $t$ is the time, $\mathcal{J} \colon \mathbb{R}^p \to \mathbb{R}$ is a loss function to be minimized (e.g., empirical loss in DL applications) as in Equation (1), assumed $C^2$ with gradient $\nabla\mathcal{J}$ and Hessian $\nabla^2\mathcal{J}$. A key aspect of Equation (DIN) that places it between first- and second order optimization is that a change of variables allows to describe it using only the gradient $\nabla\mathcal{J}$, since $\nabla^2\mathcal{J}(\theta(t))\dot{\theta}(t) = \frac{d}{dt}\nabla\mathcal{J}(\theta(t))$ (see Section 2.2 for details). This greatly reduces computational costs, as it can be discretized as a difference of gradients which does not require Hessian vector product, making it possible to design more practical algorithms, as shown in Chen & Luo (2019); Castera et al. (2021); Attouch et al. (2022).

We recover the continuous-time heavy ball system by assuming $\alpha > 0$, and removing the geometrical "damping" term in Equation (DIN) through the choice $\beta = 0$. A discrete version of this system corresponds to the Heavy Ball method (Polyak, 1964), which is at the basis of SGD solvers with momentum in deep learning (Qian, 1999; Sutskever et al., 2013). By allowing both $\alpha$ and $\beta$ to vary, we recover Nesterov acceleration (Nesterov, 1983; Su et al., 2016; Attouch et al., 2019).

**Adaptive methods.** Adaptive optimization methods, such as RMSprop (Tieleman et al., 2012) and AdaGrad (Duchi et al., 2011), use coordinate-wise scaling to stabilize the system and improve training. These methods can be modeled by continuous-time ODEs of the following form:

$$\dot{\theta}(t) + \frac{1}{\sqrt{G(t) + \epsilon}} \odot \nabla\mathcal{J}(\theta(t)) = 0, \quad t \geq 0, \tag{2}$$

where $\epsilon > 0$, $G(t) \in \mathbb{R}^p$ represents accumulated information. The scalar addition, square root, and division are understood coordinatewise and $\odot$ denotes the coordinate-wise product for vectors in $\mathbb{R}^p$. In AdaGrad or RMSprop, $G(t)$ is a gradient amplitude average of the form:

$$G(t) := \int_0^t \nabla\mathcal{J}(\theta(\tau))^2 \, d\mu_t(\tau), \tag{3}$$

for different choices of $\mu_t$ — uniform for AdaGrad and moving average for RMSprop, where the square operation applies coordinatewise. This generally improves performance, see (Duchi et al., 2011; Tieleman et al., 2012).

**Our approach.** We combine the "dynamical inertial Newton" method (DIN) from Equation (DIN) with an RMSprop adaptive gradient scaling. We thus take into account second order information for the RMSprop scaling. Computationally, this second order information is expressed using a time derivative. In discrete time, this will result in a second order like behavior with the same computational cost as gradient evaluation. The resulting continuous time ODE is:

$$\ddot{\theta}(t) + \alpha\,\dot{\theta}(t) + \beta\,\frac{d}{dt}\mathrm{RMSprop}(\mathcal{J}(\theta(t))) + \mathrm{RMSprop}(\mathcal{J}(\theta(t))) = 0 \tag{4}$$

where $\mathrm{RMSprop}(\mathcal{J}(\theta(t))) = \frac{1}{\sqrt{G(t,\theta(t))+\epsilon}} \odot \nabla\mathcal{J}(\theta(t))$.

Here, $G$ is of the form (3) with an adequate time-weight distribution $\mu_t$ corresponding to the RMSprop scaling. A discretization of this continuous time system, results in our new optimizer INNAprop, see Section 2.1. Let us emphasize that the system in (4) is the superposition of three dynamical systems: the gradient flow, the heavy ball flow, and the continuous Newton flow Alvarez D & Pérez C (1998), $\frac{d}{dt}\nabla f(\theta(t)) + \nabla f(\theta(t)) = \nabla^2 f(\theta(t))\dot{\theta}(t) + \nabla f(\theta(t)) = 0$. When adapted to the RMSprop setting it becomes $\frac{d}{dt}\mathrm{RMSprop}(\mathcal{J}(\theta(t))) + \mathrm{RMSprop}(\mathcal{J}(\theta(t))) = 0$. It is easily checked that this this system drives $\mathrm{RMSprop}(\mathcal{J}(\theta(t)))$ to zero exponentially fast as in Alvarez D & Pérez C (1998). We use the term "second order information" to reflect the fact that the underlying continuous time system features second order derivatives of the cost inducing fast convergence. Note that the resulting algorithm remains a first order method, based only on gradient oracles, this is a key aspect of our approach motivated by deep learning applications.

**Relation with existing work.** To improve the efficiency of stochastic gradient descent (SGD), two primary strategies are used: leverage local geometry for having clever directions and incorporate momentum to accelerate convergence. These approaches include accelerated methods (e.g., Nesterov's acceleration (Nesterov, 1983; Dozat, 2016), momentum SGD (Polyak, 1964; Qian, 1999; Sutskever et al., 2013), and adaptive methods (e.g., Adagrad (Duchi et al., 2011), RMSprop (Tieleman et al., 2012)), which adjust learning rates per parameter.

Adam remains the dominant optimizer with numerous variants proposed to improve its performance or better suit specific cases (Dozat, 2016; Shazeer & Stern, 2018; Reddi et al., 2019; Loshchilov & Hutter, 2017; Zhuang et al., 2020). Adafactor (Shazeer & Stern, 2018) improves memory efficiency, Lamb (You et al., 2019) adds layerwise normalization, and Lion (Chen et al., 2023b) uses sign-based momentum updates. AdEMAMix (Pagliardini et al., 2024) combines two EMAs, while Defazio et al. (Defazio et al., 2024) introduced a schedule-free method incorporating Polyak-Ruppert averaging with momentum.

One of the motivations of our work is the introduction of second order properties in the dynamics akin to Newton's method. second order optimizers are computationally expensive due to frequent Hessian computations (Gupta et al., 2018; Martens & Grosse, 2015). Their adaptation to large scale learning settings require specific developments (Jahani et al., 2021; Qian et al., 2021; Liu et al., 2023).

**Contributions.**
— Our method, INNAprop, integrates the algorithm INNA (Castera et al., 2021), which features a computationally cheap Newtonian effect with the gradient scaling mechanism of RMSprop. We improve the memory footprint of INNA by saving a memory slot. Computational costs are similar to first order methods like AdamW, making INNAprop suitable for deep learning (see Section 2.2 and Appendix B). Specific hyperparameter choices for our method allow us to recover several existing optimizers as special cases.
— We provide a continuous-time explanation of INNAprop, connecting it to second order ordinary differential equations (see Section 2 and Equation (4)). We discuss many natural possible discretizations and show that INNAprop is empirically the most efficient. We show that second order terms in space may be approximated without resorting to Hessian computations or inversions of linear systems which are both prohibitive in deep learning.
— We show through extensive experiments that INNAprop offers close performances to AdamW and sometimes significantly better in both training speed and final accuracy on benchmarks such as image classification (CIFAR-10, ImageNet) and language modeling (GPT-2) (see Section 3), for which we also consider the Muon optimizer.

We describe our algorithm and its derivation in Section 2. Hyperparameter tuning recommendations and our experimental results are provided in Section 3.

## 2 INNAprop: a second order method in space and time based on RMSprop

### 2.1 The algorithm

Algorithm 1, which blends INNA and RMSprop, is the method we used in all experiments. It includes the usual ingredients of deep-learning training: mini-batching, decoupled weight-decay, and scheduler procedure. For a simpler, "non-deep learning" version, refer to Algorithm 2 in Appendix B.

---

**Algorithm 1** Deep learning implementation of INNAprop

---

1: **Loss function:** $\mathcal{J}(\theta) = \frac{1}{n} \sum_{n=1}^{N} \mathcal{J}_n(\theta)$ for $\theta \in \mathbb{R}^p$.
2: **Learning step-sizes:** $\gamma_k := \{\text{SetLrSchedule}(k)\}_{k \in \mathbb{N}}$; $\gamma_0$ is called the initial learning rate.
3: **Hyper-parameters:** $\sigma \in [0,1]$, $\alpha \geq 0$, $\beta > \sup_{k \in \mathbb{N}} \gamma_k$, $\lambda \geq 0$, $\epsilon = 10^{-8}$.
4: **Mini-batches:** $(\mathsf{B}_k)_{k \in \mathbb{N}} \subset \{1, \ldots, N\}$ chosen with usual protocols, see e.g., Krizhevsky et al. (2012).
5: **Initialization:** time step $k \leftarrow 0$, parameter vector $\theta_0$, $v_0 = 0$, $\psi_0 = (1 - \alpha\beta)\theta_0$.
6: **for** $k = 1$ **to** K **do**
7: $\quad \boldsymbol{g}_k = \frac{1}{|\mathsf{B}_k|} \sum_{n \in \mathsf{B}_k} \nabla \mathcal{J}_n(\boldsymbol{\theta}_k)$
8: $\quad \gamma_k \leftarrow \text{SetLrSchedule}(k)$ $\qquad\qquad\qquad\qquad\qquad\qquad\qquad\qquad$ ▷ see above
9: $\quad \boldsymbol{\theta}_k \leftarrow (1 - \lambda\gamma_k)\boldsymbol{\theta}_k$ $\qquad\qquad\qquad\qquad\qquad\qquad\qquad$ ▷ decoupled weight decay
10: $\quad \boldsymbol{v}_{k+1} \leftarrow \sigma\boldsymbol{v}_k + (1-\sigma)\boldsymbol{g}_k^2$
11: $\quad \hat{\boldsymbol{v}}_{k+1} \leftarrow \boldsymbol{v}_{k+1}/(1-\sigma^k)$
12: $\quad \boldsymbol{\psi}_{k+1} \leftarrow \left(1 - \frac{\gamma_k}{\beta}\right)\boldsymbol{\psi}_k + \gamma_k\left(\frac{1}{\beta} - \alpha\right)\boldsymbol{\theta}_k$
13: $\quad \boldsymbol{\theta}_{k+1} \leftarrow \left(1 + \frac{\gamma_k(1-\alpha\beta)}{\beta-\gamma_k}\right)\boldsymbol{\theta}_k - \frac{\gamma_k}{\beta-\gamma_k}\boldsymbol{\psi}_{k+1} - \gamma_k\beta\left(\boldsymbol{g}_k/(\sqrt{\hat{\boldsymbol{v}}_{k+1}} + \epsilon)\right)$
14: **return** $\theta_{K+1}$

---

SetLrSchedule is the "scheduler" for step-sizes. It is a custom procedure for handling learning rate sequences: detailed explanations of several procedures along with the corresponding benchmarks in Section 3 may be found in Appendix D.

Observe that, for all schedulers $\gamma_k < \beta$ for $k \in \mathbb{N}$, so that INNAprop is well-posed (see line 13 in Algorithm 1).

### 2.2 Derivation of the algorithm

We start with the INNA optimizer of Castera et al. (2021) without minibatch, which is an Euler discretization of the system (DIN). Introducing, for each $k \in \mathbb{N}$ a step-size $\gamma_k$ and an estimate $g_k$ for $\nabla J(\theta_k)$, the algorithm is given in the a phase space form as as follows

$$\psi_{k+1} = \psi_k\left(1 - \frac{\gamma_k}{\beta}\right) + \gamma_k\left(\frac{1}{\beta} - \alpha\right)\theta_k, \qquad \theta_{k+1} = \theta_k\left(1 + \gamma_k\left(\frac{1}{\beta} - \alpha\right)\right) - \frac{\gamma_k}{\beta}\psi_k - \gamma_k\beta g_k. \qquad \text{(INNA)}$$

A direct modification of the gradient estimate $g_k$ allows to include the RMSprop mechanism.

$$v_{k+1} = \sigma v_k + (1-\sigma)g_k^2$$
$$\psi_{k+1} = \psi_k\left(1 - \frac{\gamma_k}{\beta}\right) + \gamma_k\left(\frac{1}{\beta} - \alpha\right)\theta_k, \qquad\qquad \text{(INNAprop-0)}$$
$$\theta_{k+1} = \theta_k\left(1 + \gamma_k\left(\frac{1}{\beta} - \alpha\right)\right) - \frac{\gamma_k}{\beta}\psi_k - \gamma_k\beta\frac{g_k}{\sqrt{v_{k+1}} + \epsilon}.$$

The final version of the recursion in Algorithm 1 is a variation of the base recursion in (INNAprop-0) with the following features:

- A bias correction step for the RMSprop term, decoupled weight decay and learning rate schedulers. These are typical in modern deep learning optimizers.

- An update form equivalent to (INNAprop-0) which allows for a naive implementation with in place updates for the variables $\theta$ and $\psi$, contrary to (INNAprop-0), see Appendix B.1 for more details.

A few remarks are in order.

**Remark 1 (On memory savings)** The formulation of INNAprop used in Algorithm 1 directly allows for in place updates, potentially saving memory. Alternatively a careful implementation of the original form of (INNAprop-0), the use of additional computational devices, such as compilers, could possibly lead to similar memory savings.

**Remark 2 (A family of algorithms indexed by $\alpha, \beta$)** INNAprop is a family of methods indexed by the hyperparameters $\alpha$ and $\beta$, with an interesting special case. For $\alpha = \beta = 1$, INNAprop with its default initialization, boils down to AdamW without momentum ($\beta_1 = 0$), see Appendix B.3 and Table 5. This correspondence is confirmed empirically in Figure 9 and Figure 8 in Appendix B.3 for a VGG and a ResNet model. Regarding the parameter $\alpha$ in Algorithm 1 line 12 and 13, it may be interpreted as having an effect that is similar to uncoupled weight decay on line 9. While this intuition makes sense, varying $\lambda$ only affects the $\theta$ variable while $\alpha$ affects both $\theta$ and $\psi$. We confirm in the experimental section that the parameter $\alpha$ plays a role which goes beyond that of a hidden uncoupled weight decay parameter.

**Remark 3 (On other possible discretizations)** It can be checked that the basic recursion (INNAprop-0) can be interpreted as an Euler discretization of the continuous dynamics (4), see Appendix B.2 for more details. We mention for completeness different attempts at discretizing the continuous time system in (4) resulting in two algorithmic schemes which were not considered for further evaluation.
(a) RMSprop with momentum (Graves, 2013) is obtained by a discretization of the heavy ball continuous time system, using an RMSprop proxy instead of the gradient. It would be natural to proceed similarly in our case with the system (4) instead of the heavy ball system. This leads to a different method (see Appendix C.1), which appears to be numerically unstable on early experiments (see Figure 10 for an illustration).
(c) For Adam, the RMSprop mechanism is applied not in the momentum term but only in the $\theta$ update, see Table 5. Doing similarly with the system (4) leads to a third method, detailed in Appendix C.2. We did not push the investigations for this method as it essentially amounts to run the NAdam algorithm (Dozat, 2016) for different hyper-parameter choices (see Appendix C.2 for details).

## 3   Empirical evaluation of INNAprop

We conduct extensive comparisons of the proposed algorithm and the AdamW optimizer, which is dominantly used in image classification (Chen et al., 2018; Zhuang et al., 2020; Touvron et al., 2021; Mishchenko & Defazio, 2023) and language modeling tasks (Brown et al., 2020; Hu et al., 2021; Liu et al., 2023). Hyperparameter tuning (Sivaprasad et al., 2020) is a crucial issue for this comparison, and we start with this. As a general rule, we strive to choose the hyperparameters that give a strong baseline for AdamW (based on literature or using grid search). In particular, most of our experiments use the AdamW optimizer [2] with its default settings as defined in widely-used libraries (Paszke et al., 2019; Bradbury et al., 2018; Abadi et al., 2016): $\beta_1 = 0.9$, $\beta_2 = 0.999$, $\lambda = 0.01$ and $\epsilon = 10^{-8}$. For INNAprop, unless otherwise specified, the default settings for the RMSprop component align with those of AdamW: $\sigma = 0.999$ and $\epsilon = 10^{-8}$.

The INNAprop method and the AdamW optimizer involve different classes of hyperparameters; some of them are common to both algorithms, and some are specific. Our hyperparameter tuning strategy for both algorithms is summarized in Table 1, AdamW is favored whenever possible. We emphasize that AdamW, which is regarded as one of the if not the state-of-the-art algorithm, has benefited from a long history of extensive testing by a large research community. Favoring AdamW through robust, community-validated setups provides this method with a considerable advantage. It should also be noted, that the notion of 'state of the art' does not necessarily correspond to an absolute optimum, as performance may depend on specific implementations and hardware configurations which is not an aspect we take into account presently.

---

[2] https://pytorch.org/docs/stable/generated/torch.optim.AdamW.html

Table 1: Hyperparameter tuning strategy for INNAprop and AdamW.

| Parameters | AdamW tuning | INNAprop tuning | Advantage |
|---|:---:|:---:|:---:|
| Learning rate | Literature or grid search tuning | Reused from AdamW | AdamW |
| Step size scheduler | Literature | Reused from AdamW | N/A |
| Weight decay | Literature or grid search tuning | Reused from AdamW | AdamW |
| RMSprop parameter | Default or literature | Reused from AdamW | AdamW |
| Inertial parameters $(\alpha, \beta)$ | N/A | Tuned on CIFAR-10 | N/A |

We begin this section with the tuning of parameters $\alpha, \beta$ for INNAprop on CIFAR10 with VGG and ResNet architectures and then use these parameters on larger datasets and models. As explained previously, we use as much as possible the step size scheduler and weight decay settings reported in the literature for the AdamW optimizer, which we believe to be well-adjusted and provide adequate baselines for each experiment. *These are then reused for INNAprop, whence, with this protocol, we merely tune $\alpha$ and $\beta$.* This approach is guided by constrained computational resources. It is, however, very interesting to look for INNAprop's peak performance with extensive tuning.

**Remark 4 (Choice of baselines)** Although SGD with momentum is a classical baseline for CNNs, recent works such as Lion (Chen et al., 2023b) and Prodigy (Mishchenko & Defazio, 2023) show that AdamW can outperform it when properly tuned. In light of this, we adopt AdamW as our primary baseline. We also do not include very recent optimizers such as Muon (Jordan et al., 2024), Scion (Pethick et al., 2025), or SchedulerFree (Defazio et al., 2024) as they currently lack established benchmark protocols and reliable hyperparameter defaults. A comprehensive comparison would require extensive tuning, which is beyond the scope of this work.

### 3.1 Tuning INNAprop on CIFAR-10 with VGG11 and ResNet18

**Hyperparameter tuning:** We consider VGG11 (Simonyan & Zisserman, 2014) and ResNet18 (He et al., 2016) models trained on CIFAR10 (Krizhevsky & Hinton, 2010). We fix a cosine scheduler where $T_{\max} = 200$, as recommended for AdamW, and $\gamma_{\min} = 0$ (see Appendix D for more details) and consider two weight decay parameters $\lambda = 0$ or $\lambda = 0.01$ (defaut value for AdamW). We tune the initial learning rate $\gamma_0$ only for AdamW. We find $\gamma_0 = 10^{-3}$, which is also the baseline value reported for AdamW in this experiment (see Appendix E).

For INNAprop, we tune only $\alpha$ and $\beta$ using $\gamma_0 = 10^{-3}$ from AdamW. Using `optuna` (Akiba et al., 2019), we perform a grid search over $(\alpha, \beta) \in \{0.1, 0.5, 0.9, \dots, 3.5, 4.0\}$. In Figure 1, we detail the obtained training loss and test accuracy for various $(\alpha, \beta)$ configurations over short training durations (20 epochs) and long training durations (200 epochs) for VGG11. Our criteria (short and long training duration) are chosen to find parameters $(\alpha, \beta)$ that provide a rapid decrease in training loss in the early stages and the best test accuracy for long training duration.

We observe that $(\alpha, \beta) = (0.1, 0.9)$ gives one of the best training losses for short durations, while $(\alpha, \beta) = (2.0, 2.0)$ works well for longer durations. Similar results for VGG11 and ResNet18 with and without weight decay are in Appendix F.4.

**Validation and comparison with AdamW:** We validate this observation by repeating the experiment with 8 random seeds and comparing it to AdamW under the same settings with three common architectures: VGG11 Simonyan & Zisserman (2014), ResNet-18 He et al. (2016), both used for tuning, and an additional DenseNet121 architecture Huang et al. (2017) to verify that our choices transfer to different models. We show in Figure 13 that the batch size has a rather limited influence.

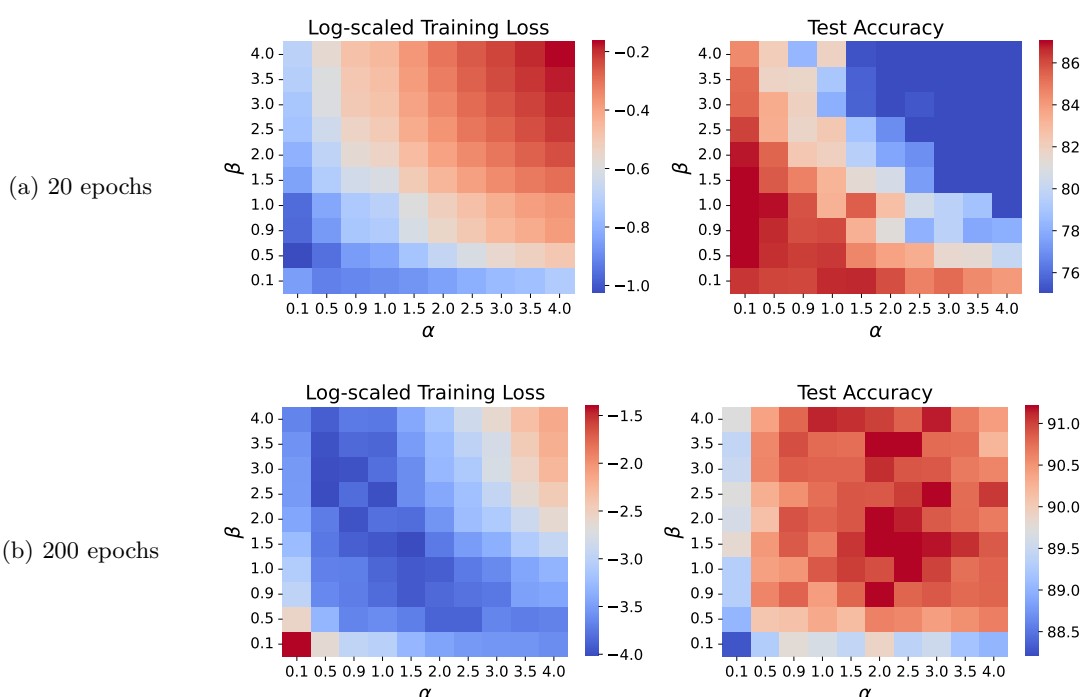

Figure 1: Log-scale training loss and test accuracies for hyperparameters $(\alpha, \beta)$ with VGG11 on CIFAR10 at 20 and 200 epochs. Optimal learning rate $\gamma_0 = 10^{-3}$ and weight decay $\lambda = 0.01$, with one random seed.

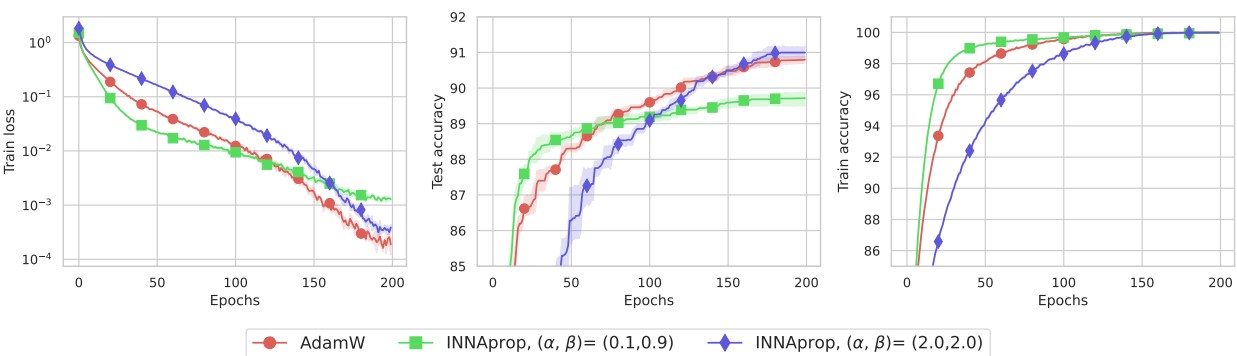

Figure 2: Training VGG11 on CIFAR10 with 8 random seeds.

In Figure 2 (refer to Appendix F for additional experiments with the ResNet-18), we observe that with $(\alpha, \beta) = (0.1, 0.9)$, INNAprop improves training loss and test accuracy quickly by the 100th epoch, achieving the highest training accuracy. With $(\alpha, \beta) = (2.0, 2.0)$, INNAprop trains more slowly, but achieves a higher final test accuracy. This is aligned with the experiments described in Figure 1. In Table 2, we compare the performance of different networks on CIFAR-10 using INNAprop and AdamW optimizers. INNAprop outperforms AdamW for this benchmark.

Let us now use these tunings for large-scale models.

## 3.2 Extensive experiments on large-scale vision models

Let us present our experiments on large-scale vision benchmarks with the hyperparameters of Section 3.1.

Table 2: Test accuracy (%) of ResNet-18, VGG11, and DenseNet121 on CIFAR-10. AdamW weight decay & learning rate are optimized. Results are averaged over eight runs.

| Model | Optimizer | Test accuracy |
|---|---|---|
| | Training on CIFAR-10 over 200 epochs | |
| ResNet18 | AdamW | 91.14 |
| | INNAprop ($\alpha = 2.0, \beta = 2.0$) | **91.58** |
| VGG11 | AdamW | 90.79 |
| | INNAprop ($\alpha = 2.0, \beta = 2.0$) | **90.99** |
| DenseNet121 | AdamW | 86.19 |
| | INNAprop ($\alpha = 0.1, \beta = 0.9$) | **86.91** |

**Resnets on ImageNet:** We consider the ImageNet-1k benchmark (Krizhevsky et al., 2012), training for 90 epochs on ResNet-18 and ResNet-50 (He et al., 2016) with a mini-batch size of 256, following Chen et al. (2023b); Zhuang et al. (2020). We used a state-of-the-art cosine scheduler for AdamW, initialized with a learning rate of $\gamma_0 = 10^{-3}$, as reported in Chen et al. (2023b); Zhuang et al. (2020); Chen et al. (2018), and applied it to INNAprop. The weight decay of AdamW was set to $\lambda = 0.01$ for the ResNet18, instead of $\lambda = 0.05$ reported in Zhuang et al. (2020); Chen et al. (2018) because it improved the test accuracy from 67.93 to 69.43. The results of the ResNet18 experiment are presented in Figure 16 in Appendix F. The figure shows, despite the lack of hypertuning, that our algorithm with $(\alpha, \beta) = (0.1, 0.9)$ outperforms AdamW in test accuracy (70.12 vs 69.34). However, observe that the training loss decreases faster initially but slows down towards the end of training.

For the ResNet50, we kept the value $\lambda = 0.1$ as reported in Zhuang et al. (2020); Chen et al. (2018). For INNAprop, we tried two weight decay values $\{0.1, 0.01\}$ and selected $\lambda = 0.01$ as it resulted in a faster decrease in training loss. We report the results in Figure 3, illustrating the advantage of INNAprop. As noted in Section 3.1, INNAprop with $(\alpha, \beta) = (0.1, 0.9)$ trains faster but has a lower test accuracy compared to AdamW/INNAprop with $(\alpha, \beta) = (2.0, 2.0)$. For $(\alpha, \beta) = (2.0, 2.0)$, the loss decrease is similar to AdamW, with no clear advantage for either method. This obviously suggests developing scheduling strategies for the damping parameters $(\alpha, \beta)$. This requires a much more computation-intensive tuning, far beyond the numerical resources used in the current work. In Table 3, we present the performance of INNAprop achieved using minimal hyperparameter tuning, as explained in Table 1

**Vision transformer (ViT) on ImageNet:** We performed the same protocol (state-of-the-art AdamW parameters) with a ViT-B/32 architecture over 300 epochs with a mini-batch size of 1024, following Defazio & Mishchenko (2023); Mishchenko & Defazio (2023). We used a cosine scheduler with a linear warmup (30 epochs) and the initial learning rate and weight decay from Defazio & Mishchenko (2023). We confirm in Figure 18, that this is a favorable weight decay choice for AdamW on this problem. For INNAprop, we tested weight decay values of $\{0.1, 0.01\}$, selecting $\lambda = 0.1$ for better test accuracy. Results in Figure 3 show the advantage of INNAprop. For faster convergence using INNAprop $(0.1, 0.9)$, we recommend a weight decay of $\lambda = 0.01$ (see Figure 17 in Appendix F).

In the ImageNet experiments, we evaluated INNAprop for rapid early training and optimal final test accuracy without tuning $(\gamma_0, \alpha, \beta)$. For ViT-B/32 with $\lambda = 0.1$, INNAprop achieved higher training loss and a similar final test accuracy compared to AdamW (75.23 vs. 75.02). Note that the choice of weight decay is tuned for AdamW on this problem (see Figure 18 in Appendix F) confirming the comment in Remark 2 regarding the role of $\lambda$ and $\alpha$.

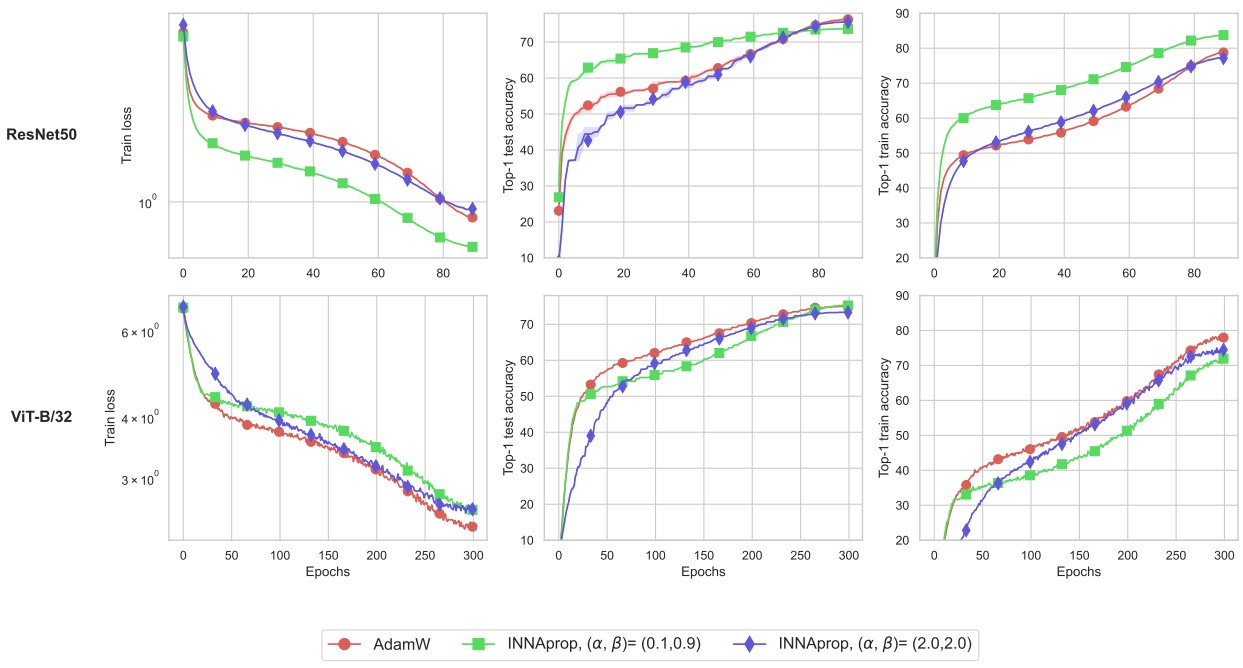

Figure 3: Training a ResNet50 (top) and ViT-B/32 (bottom) on ImageNet with 3 random seeds.

Table 3: Top-1 and Top-5 accuracy (%) of ResNet-18, ResNet-50, and ViT-B/32 on ImageNet. Results are averaged from three runs for ResNets and one run for ViT-B/32.

| Model | Optimizer | Top-1 acc | Top-5 acc |
|---|---|---|---|
| \multicolumn{4}{c}{Train from scratch on ImageNet} | | | |
| ResNet18 | AdamW | 69.34 | 88.71 |
| | INNAprop ($\alpha = 0.1, \beta = 0.9$) | **70.12** | **89.21** |
| ResNet50 | AdamW | 76.33 | 93.04 |
| | INNAprop ($\alpha = 1.0, \beta = 1.0$) | **76.43** | **93.15** |
| ViT-B/32 | AdamW | 75.02 | 91.52 |
| | INNAprop ($\alpha = 0.1, \beta = 0.9$) | **75.23** | **91.77** |

**Fintetuning VGG11 and ResNet18 models on Food101:** We fine-tuned ResNet-18 and VGG-11 models on the Food101 dataset (Bossard et al., 2014) for 20 epochs, using pre-trained models on ImageNet-1k. Since weight decay and learning rate values for AdamW were not found in the literature, we chose the default AdamW weight decay value, $\lambda = 0.01$. We used a cosine scheduler and tried one run for each initial learning rate value in $\{10^{-5}, 5 \times 10^{-5}, 10^{-4}, 5 \times 10^{-4}, 10^{-3}\}$. The best result for AdamW was obtained for $\gamma_0 = 10^{-4}$, and we kept the same setting for INNAprop.

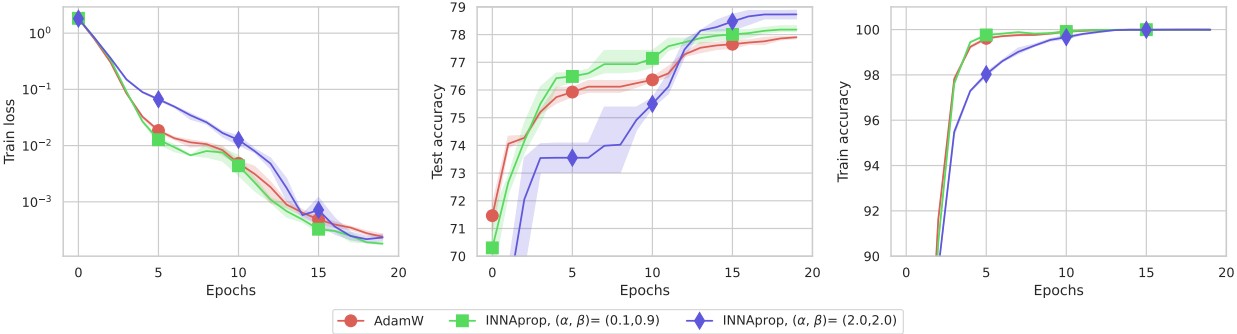

Figure 4: Finetuning a VGG11 on Food101. Left: train loss, middle: test accuracy (%), right: train accuracy (%). Qualitatively similar results for ResNet18 are in Figure 14 in Appendix F. 3 random seeds.

See for this Figure 4, where INNAprop performs no worse than AdamW on three random seeds.

**Conclusion and recommendations for image classification:** Tuning $(\alpha, \beta)$ significantly impacts training. Based on heatmaps in Section 3.1 and figures in Section 3.2, we recommend using $\alpha = 0.1$ and $\beta \in [0.5, 1.5]$ for shorter training (e.g., fine-tuning). For longer training, $\alpha, \beta \geq 1$ is preferable. In both cases, our algorithm either matches or outperforms AdamW.

### 3.3 Pre-training and fine-tuning GPT2

We present experimental results on LLMs using the hyperparameters selected in Section 3.1.

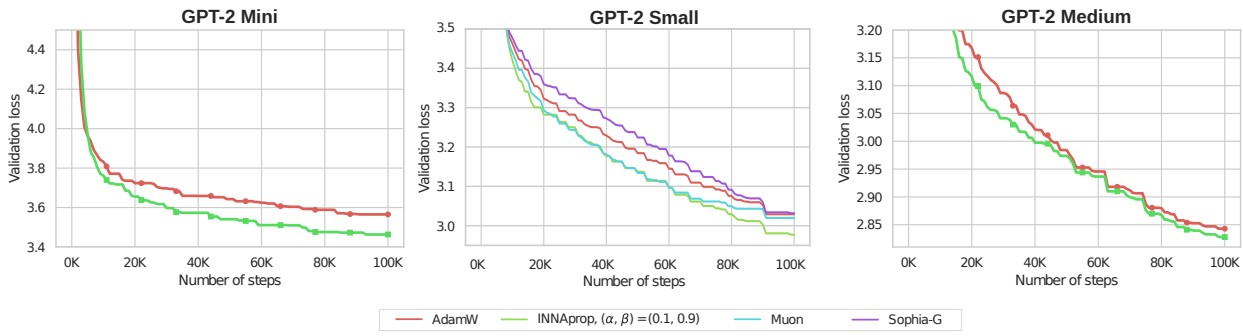

Figure 5: GPT-2 training from scratch on OpenWebText (Sophia-G unstable on mini and medium). The implementation of the Muon optimizer is adapted from the Modded-NanoGPT, see main text.

**Training GPT-2 from scratch:** We trained various GPT-2 transformer models from scratch (Radford et al., 2019) using the nanoGPT repository[3] on the OpenWebText dataset Gokaslan & Cohen (2019). For all models, gradients are clipped to a norm of 1, following Brown et al. (2020). We use AdamW with hyperparameters from the literature (Liu et al., 2023; Brown et al., 2020), the standard configuration for LLM pre-training. The reported RMSprop parameter $\beta_2 = 0.95$ is different from AdamW's default (0.999),

---

[3]https://github.com/karpathy/nanoGPT

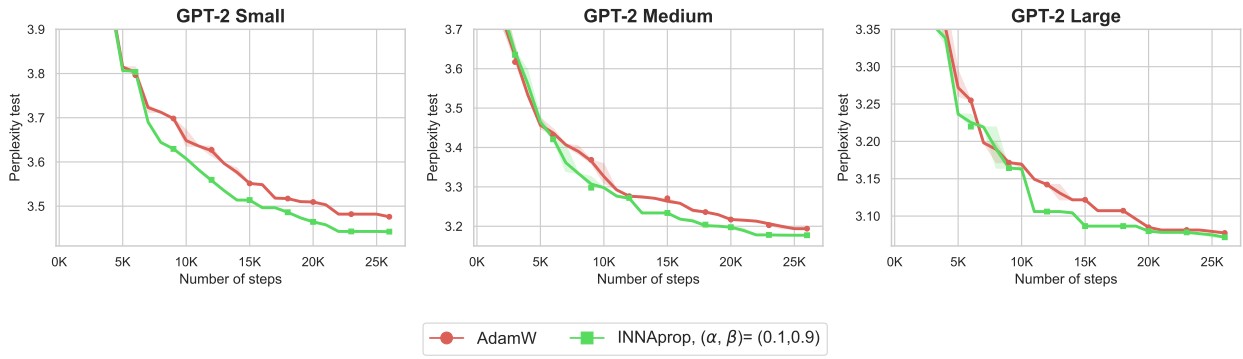

Figure 6: Perplexity test with GPT-2 E2E Dataset with LoRA finetuning on five epochs. Three random seeds.

the weight decay is $\lambda = 0.1$ and $\gamma_0$ depending on the network size. For example, GPT-2 small works with an initial learning rate $\gamma_0 = 6 \times 10^{-4}$. For INNAprop, we keep the same values for $\lambda$ and $\gamma_0$ as AdamW, and use the RMSprop parameter $\sigma = 0.99$ (corresponding to $\beta_2$ for AdamW), which provides the best results among values $\{0.9, 0.95, 0.99\}$ on GPT-2 mini. We use this setting for all our GPT-2 experiments with $(\alpha, \beta) = (0.1, 0.9)$. The results are in Figure 5. INNAprop leads to a faster decrease in validation loss during the early stages compared to the baseline for GPT-2 models of Mini (30M), Small (125M), and Medium (355M) sizes. Its performance could be further improved with a more thorough tuning of all hyperparameters $(\alpha, \beta, \sigma, \lambda)$. For GPT-2 small, we also include a comparison with two recent algorithms. Sophia-G, using the hyperparameters provided in the literature [4] (Liu et al., 2023). The Muon optimizer Jordan et al. (2024), based on the Modded-NanoGPT repository codebase[5], which is part of a collaborative competition for fast GPT-2 training on a 8xH100 cluster. For Muon, we took the minibatch size, schedule and weight decay from this repository. We adapted the number of GPUs to fit our cluster, also impacting the total number of training tokens. We also used the same nanoGPT architecture as we used for AdamW and INNAprop and the OpenWebText dataset.

**Comparision with INNA** As a sanity check, we evaluate INNA on GPT-2 Mini and compare it to INNAprop and AdamW. Following Castera et al. (2021), we used the recommended hyperparameters $(\alpha, \beta) = (0.5, 0.1)$ and tested learning rates $\{1e-4, 1e-3, 1e-2, 1e-1\}$, selecting $\gamma_0 = 0.1$ as the best. Figure 7 shows that INNAprop and AdamW outperform INNA in both convergence speed and final validation loss.

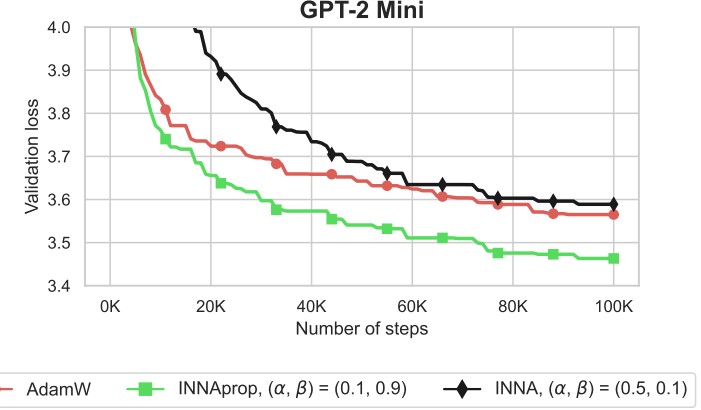

Figure 7: Validation loss comparison during GPT-2 mini training from scratch on the OpenWebText dataset.

---

[4]https://github.com/Liuhong99/Sophia

[5]https://github.com/KellerJordan/modded-nanogpt

**Fine-tune GPT-2 with LoRA:** Using LoRA (Hu et al., 2021), we fine-tune the same GPT-2 models on the E2E dataset Novikova et al. (2017), consisting of roughly 42000 training examples, 4600 test examples from the restauration domain. We compare AdamW and INNAprop for 5 epochs, as recommended in Hu et al. (2021). We use state-of-the-art parameters for AdamW: linear learning rate schedule, the recommended mini-batch size, and the RMSprop parameter ($\beta_2 = \sigma = 0.999$), these are listed in Table 11 in Hu et al. (2021). The same parameters are *reused without further tuning* for INNAprop, yet the latter outperforms AdamW. The results are displayed in Figure 6 and Table 4, where we see the perplexity mean result on 3 random seeds. INNAprop with $(\alpha, \beta) = (0.1, 0.9)$ achieves lower perplexity loss compared to AdamW across all GPT-2 fine-tuning experiments.

We summarize the performance of our algorithm on LLMs (see Table 4); we emphasize the capabilities of INNAprop compared to AdamW in the context of early training where gains are considerable.

Table 4: Comparison of GPT-2 training on OpenWebText (validation loss) and LoRA fine-tuning on E2E (perplexity).

| Model | AdamW | INNAprop | Steps to match AdamW |
|---|---|---|---|
| **GPT-2 from scratch (best validation loss)** | | | |
| GPT-2 mini | 3.57 | **3.47** | 51,000 (**1.96× faster**) |
| GPT-2 small | 3.03 | **2.98** | 79,000 (**1.26× faster**) |
| GPT-2 medium | 2.85 | **2.82** | 83,000 (**1.2× faster**) |
| **GPT-2 with LoRA (best perplexity test)** | | | |
| GPT-2 small | 3.48 | **3.44** | 19,000 (**1.31× faster**) |
| GPT-2 medium | 3.20 | **3.17** | 20,000 (**1.25× faster**) |
| GPT-2 large | 3.09 | **3.06** | 20,000 (**1.25× faster**) |

## 4 Conclusion

INNAprop leverages second order geometric information in a cheap manner while maintaining memory and computational footprints similar to AdamW. Experiments on text modeling and image classification show that INNAprop consistently matches or exceeds AdamW's performance.

We systematically favored AdamW by adopting the community-recommended hyperparameters, including schedulers, learning rates, and weight decay, which result from extensive empirical efforts to optimize the method's performance across diverse tasks. For INNAprop, hyperparameter tuning for friction parameters $(\alpha, \beta)$ was conducted using a grid search on CIFAR-10, see Figure 21. We found that the resulting method performs very well across a variety of settings (both in models and datasets), and outperforms AdamW for certain tasks such as GPT-2 training, without any additional hyperparameter tuning. We believe that further experimental studies with INNAprop could greatly improve its efficiency.

For language models, $(\alpha, \beta) = (0.1, 0.9)$ works well for both pre-training and fine-tuning. In image classification, $(\alpha, \beta) = (0.1, 0.9)$ speeds up early training, while $(\alpha, \beta) = (2.0, 2.0)$ improves test accuracy in long runs and during fine-tuning.

Overall, INNAprop shows strong performance across different benchmarks and model scales, making it a promising option for training large models. Future research will focus on the design of schedulers for hyperparameters $\alpha$ and $\beta$.

## Acknowledgements

Access to MesoNET resources in Toulouse was granted under allocation m23038. The authors thank Nicolas Renon and Calmip support team for their help in setting up the experiments. The authors thank AI Interdisciplinary Institute ANITI funding, through the French "Investments for the Future – PIA3" program under the grant agreement ANR-23-IACL-0002, Air Force Office of Scientific Research, Air Force Material Command, USAF, under grant numbers FA8655-22-1-7012, ANR MAD ANR-24-CE23-1529, ANR Regulia. JB and EP thank TSE-P and acknowledge support from ANR Chess, grant ANR-17-EURE-0010. EP acknowledges support from IUF and ANR Bonsai, grant ANR-23-CE23-0012-01.

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

This is the appendix for "A second-order-like optimizer with adaptive gradient scaling for deep learning".

## Contents

# A Reminder on optimization algorithms

Considering the unconstrained minimization of $\mathcal{J}$ (see Equation (1)) and using the shorthand $\nabla \mathcal{J}(\theta_k) = g_k$, we outline in the table below the structure of several well-known optimizers.

Table 5: SGD is due to (Robbins & Monro, 1951), Momentum to (Polyak, 1964), Nesterov to (Nesterov, 1983), RMSprop + Momentum to (Graves, 2013), Adam to (Kingma & Ba, 2014), NAdam to (Dozat, 2016) and INNA to (Castera et al., 2021).

**SGD($\gamma_k$)**

$$\theta_{k+1} = \theta_k - \gamma_k g_k$$

**Adam($\gamma_k, \beta_1, \beta_2, \epsilon$)**

$$m_0 = 0, v_0 = 0$$
$$m_{k+1} = \beta_1 m_k + (1 - \beta_1)g_k$$
$$v_{k+1} = \beta_2 v_k + (1 - \beta_2)g_k^2$$
$$\theta_{k+1} = \theta_k - \gamma_k \frac{m_{k+1}}{\sqrt{v_{k+1}} + \epsilon}$$

**NAdam($\gamma_k, \psi, \beta_1, \beta_2, \epsilon$)**

$$m_0 = 0, v_0 = 0$$
$$\mu_k = \beta_1(1 - \frac{1}{2}0.96^{k\psi})$$
$$m_{k+1} = \beta_1 m_k + (1 - \beta_1)g_k$$
$$v_{k+1} = \beta_2 v_k + (1 - \beta_2)g_k^2$$
$$\theta_{k+1} = \theta_k - \gamma_k \frac{\mu_{k+1} m_{k+1} + (1 - \mu_k)g_k}{\sqrt{v_{k+1}} + \epsilon}$$

**DINAdam($\gamma_t, \sigma_1, \sigma_2, \alpha, \beta, \epsilon$)**

$$m_0 = 0, v_0 = 0$$
$$m_{k+1} = \sigma_1 m_k + (1 - \sigma_1)(1 - \beta\alpha)g_k$$
$$v_{k+1} = \sigma_2 v_k + (1 - \sigma_2)g_k^2$$
$$\theta_{k+1} = \theta_k - \gamma_k \frac{m_{k+1} + \alpha\beta g_k}{\sqrt{v_{t+1}} + \epsilon}$$

**SGD + Momentum($\gamma_k, \beta_1$)**

$$v_0 = 0$$
$$v_{k+1} = \beta_1 v_k + (1 - \beta_1)g_k$$
$$\theta_{k+1} = \theta_k - \gamma_k v_{k+1}$$

**RMSprop + Momentum($\gamma_k, \beta_1, \beta_2, \epsilon$)**

$$v_0 = 1, m_0 = 0$$
$$v_{k+1} = \beta_2 v_k + (1 - \beta_2)g_k^2$$
$$m_{k+1} = \beta_1 m_k + \frac{g_k}{\sqrt{v_{k+1}} + \epsilon}$$
$$\theta_{k+1} = \theta_k - \gamma_k m_{k+1}$$

**INNA($\gamma_k, \alpha, \beta$)**

$$\psi_0 = (1 - \alpha\beta)\theta_0$$
$$\psi_{k+1} = \psi_k + \gamma_k \left( (\frac{1}{\beta} - \alpha)\theta_k - \frac{1}{\beta}\psi_k \right)$$
$$\theta_{k+1} = \theta_k + \gamma_k \left( (\frac{1}{\beta} - \alpha)\theta_k - \frac{1}{\beta}\psi_k - \beta g_k \right)$$

**INNAprop($\gamma_k, \alpha, \beta, \sigma_2$)**

$$\psi_0 = (1 - \alpha\beta)\theta_0$$
$$v_{k+1} = \sigma v_k + (1 - \sigma)g_k^2$$
$$\psi_{k+1} = \psi_k \left( 1 - \frac{\gamma_k}{\beta} \right) + \gamma_k \left( \frac{1}{\beta} - \alpha \right) \theta_k$$
$$\theta_{k+1} = \left( 1 + \frac{\gamma_k(1 - \beta\alpha)}{\beta - \gamma_k} \right) \theta_k - \frac{\gamma_k}{\beta - \gamma_k}\psi_{k+1} - \gamma_k\beta \frac{g_k}{\sqrt{v_{k+1}} + \epsilon}$$

## B Additional details on the derivation of INNAprop

### B.1 An equivalent basic update

Starting again from the basic recursion (INNAprop-0),

$$v_{k+1} = \sigma v_k + (1-\sigma)g_k^2$$
$$\psi_{k+1} = \psi_k\left(1 - \frac{\gamma_k}{\beta}\right) + \gamma_k\left(\frac{1}{\beta} - \alpha\right)\theta_k,$$
$$\theta_{k+1} = \theta_k\left(1 + \gamma_k\left(\frac{1}{\beta} - \alpha\right)\right) - \frac{\gamma_k}{\beta}\psi_k - \gamma_k\beta\frac{g_k}{\sqrt{v_{k+1}} + \epsilon},$$

we obtain an equivalent rewrite of the update starting with the following equivalent relation between $\psi_k$ and $\psi_{k+1}$

$$\psi_{k+1} = \psi_k\left(1 - \frac{\gamma_k}{\beta}\right) + \gamma_k\left(\frac{1}{\beta} - \alpha\right)\theta_k, \tag{5}$$
$$\Leftrightarrow \quad \psi_k = \frac{\beta}{\beta - \gamma_k}\left(\psi_{k+1} - \gamma_k\left(\frac{1}{\beta} - \alpha\right)\theta_k\right).$$

This allows to obtain an equivalent update for the $\theta$ variable as follows

$$
\begin{aligned}
\theta_{k+1} &= \theta_k\left(1 + \gamma_k\left(\frac{1}{\beta} - \alpha\right)\right) - \frac{\gamma_k}{\beta}\psi_k - \gamma_k\beta\frac{g_k}{\sqrt{v_{k+1}} + \epsilon} \\
&= \theta_k + \gamma_k\left(\frac{1}{\beta} - \alpha\right)\theta_k - \frac{\gamma_k}{\beta - \gamma_k}\psi_{k+1} + \frac{\gamma_k}{\beta - \gamma_k}\gamma_k\left(\frac{1}{\beta} - \alpha\right)\theta_k - \gamma_k\beta\frac{g_k}{\sqrt{v_{k+1}} + \epsilon} \\
&= \theta_k + \gamma_k\left(1 + \frac{\gamma_k}{\beta - \gamma_k}\right)\left(\frac{1}{\beta} - \alpha\right)\theta_k - \frac{\gamma_k}{\beta - \gamma_k}\psi_{k+1} - \gamma_k\beta\frac{g_k}{\sqrt{v_{k+1}} + \epsilon} \\
&= \left(1 + \frac{\gamma_k(1 - \beta\alpha)}{\beta - \gamma_k}\right)\theta_k - \frac{\gamma_k}{\beta - \gamma_k}\psi_{k+1} - \gamma_k\beta\frac{g_k}{\sqrt{v_{k+1}} + \epsilon}
\end{aligned} \tag{6}
$$

This results in the following equivalent basic algorithm. This is the form we use in Algorithm 1 because it allows for a direct naive implementation using in place updates for both variables $\psi$ and $\theta$.

---

**Algorithm 2** INNAprop

1: **Objective function:** $\mathcal{J}(\theta)$ for $\theta \in \mathbb{R}^p$.
2: **Constant step-size:** $\gamma > 0$
3: **Hyper-parameters:** $\sigma \in [0,1]$, $\alpha \geq 0$, $\beta > \gamma$, $\epsilon = 10^{-8}$.
4: **Initialization:** $\theta_0$, $v_0 = 0$, $\psi_0 = (1 - \alpha\beta)\theta_0$.
5: **for** $k = 1$ **to** K **do**
6:     $g_k = \nabla\mathcal{J}(\theta_k)$
7:     $v_{k+1} \leftarrow \sigma v_k + (1-\sigma)g_k^2$
8:     $\psi_{k+1} \leftarrow \left(1 - \frac{\gamma}{\beta}\right)\psi_k + \gamma\left(\frac{1}{\beta} - \alpha\right)\theta_k$
9:     $\theta_{k+1} \leftarrow \left(1 + \frac{\gamma(1-\alpha\beta)}{\beta-\gamma}\right)\theta_k - \frac{\gamma}{\beta-\gamma}\psi_{k+1} - \gamma\beta\frac{g_k}{\sqrt{v_{k+1}}+\epsilon}$
10: **return** $\theta_{K+1}$

---

### B.2 Direct derivation from the autonomous continuous time system

Let us recall the continuous time system in (4)

$$\ddot{\theta}(t) + \alpha\,\dot{\theta}(t) + \beta\,\frac{d}{dt}\text{RMSprop}(\mathcal{J}(\theta(t))) + \text{RMSprop}(\mathcal{J}(\theta(t))) = 0 \tag{7}$$

$$\text{RMSprop}(\mathcal{J}(\theta(t))) = \frac{1}{\sqrt{G(t)+\epsilon}} \odot \nabla\mathcal{J}(\theta(t))$$

$$G(t) = \int_0^t \nabla\mathcal{J}(\theta(\tau))^2 \, d\mu_t(\tau)$$

The purpose of this section is to verify that the recursion in (INNAprop-0) can be understood as a direct Euler discretization of the system (7) for a certain probability measure $\mu_t$ over $\mathbb{R}_+$ as a time space. We will consider here a constant step size $\gamma$ for simplicity.

Let us first consider the dynamics on $G$. Set $d\mu_t(\tau) = b\exp(b(\tau-t))$ for some $b > 0$ to be precised latter. We note that $\frac{d}{dt}d\mu_t(\tau) = -bd\mu_t(\tau)$. The dynamics in $G$ can therefore be equivalently expressed as

$$\frac{d}{dt}G(t) = b\nabla\mathcal{J}(\theta(t))^2 + \int_0^t \nabla\mathcal{J}(\theta(\tau))^2 \frac{d}{dt}d\mu_t(\tau)$$

$$= b\nabla\mathcal{J}(\theta(t))^2 - b\int_0^t \nabla\mathcal{J}(\theta(\tau))^2 \, d\mu_t(\tau)$$

$$= b(\nabla\mathcal{J}(\theta(t))^2 - G(t)).$$

In (INNAprop-0), in principle, the step size $\gamma$ is small and $\sigma$ is close to 1 (for example in ADAM's default parameters). We will consider the same asymptotic regime as in (Barakat & Bianchi, 2021, Assumption 2.4) and set $b = \frac{1-\sigma}{\gamma}$. Now assuming that $\theta_k$ is given a direct Euler discretization with step size $\gamma$ gives

$$\frac{v_{k+1} - v_k}{\gamma} = \frac{1-\sigma}{\gamma}(\mathcal{J}(\theta_k)^2 - v_k)$$

which is the first line of the recursion (INNAprop-0).

The rest of the argument is similar to the derivation of the INNA algorithm in Castera et al. (2021) based on a phase space representation of the (DIN) system Alvarez et al. (2002). Introducing

$$\psi(t) = -\beta\dot{\theta}(t) - \beta^2\text{RMSprop}(\mathcal{J}(\theta(t))) - (\alpha\beta - 1)\theta(t),$$

the system (7) can be equivalently written as

$$\dot{\psi}(t) = -\frac{\psi(t)}{\beta} + \left(\frac{1}{\beta} - \alpha\right)\theta(t)$$

$$\dot{\theta}(t) = -\frac{\psi(t)}{\beta} + \left(\frac{1}{\beta} - \alpha\right)\theta(t) - \beta\text{RMSprop}(\mathcal{J}(\theta(t)))$$

and the last two lines of the recursion in (INNAprop-0) can be seen as a direct Euler discretization of this system.

### B.3 Equivalence between a special case of INNAprop and Adam without momentum

In this section, we demonstrate that INNAprop with $\alpha = 1$ and $\beta = 1$ is equivalent to Adam (Kingma & Ba, 2014) without momentum ($\beta_1 = 0$). To illustrate this, we analyze the update rules of both algorithms. We assume that the RMSprop parameter $\beta_2$ (for Adam) and $\sigma$ (for INNAprop) are equal. Starting with INNAprop, we initialize $\psi_0 = (1 - \alpha\beta)\theta_0$. For $\alpha = 1$ and $\beta = 1$, this simplifies to $\psi_0 = 0$. The update for $\psi$ becomes:

$$\psi_{k+1} = \left(1 - \frac{\gamma}{\beta}\right)\psi_k + \gamma\left(\frac{1}{\beta} - \alpha\right)\theta_k = (1-\gamma)\psi_k$$

Given that $\psi_0 = 0$, it follows that $\psi_k = 0$ for all $k$. The parameter update rule for INNAprop is:

$$\theta_{k+1} = \left(1 + \frac{\gamma(1 - \alpha\beta)}{\beta - \gamma}\right)\theta_k - \frac{\gamma}{\beta - \gamma}\psi_{k+1} - \gamma\beta\frac{g_k}{\sqrt{v_{k+1}} + \epsilon}$$

Replacing $\alpha = 1$, $\beta = 1$, and $\psi_k = 0$, we get:

$$\theta_{k+1} = \theta_k - \gamma\frac{g_k}{\sqrt{v_{k+1}} + \epsilon}$$

Here, $g_k$ is the gradient, and $v_{k+1}$ is the exponential moving average of the squared gradients:

$$v_{k+1} = \sigma v_k + (1 - \sigma)g_k^2$$

The Adam optimizer uses two moving averages, $m_k$ (momentum term) and $v_k$ (squared gradients):

$$m_k = \beta_1 m_{k-1} + (1 - \beta_1)g_k$$
$$v_k = \sigma v_{k-1} + (1 - \sigma)g_k^2$$

Setting $\beta_1 = 0$, the momentum term $m_k$ simplifies to $m_k = g_k$. The update rule becomes:

$$\theta_{k+1} = \theta_k - \gamma\frac{g_k}{\sqrt{v_k} + \epsilon}$$

This matches the form of Adam's update rule without the momentum term, confirming that INNAprop with $\alpha = 1$ and $\beta = 1$ is equivalent to Adam with $\beta_1 = 0$.

---

**Algorithm 3** INNAprop with $(\alpha, \beta) = (1, 1)$

---

1: **Objective function:** $\mathcal{J}(\theta)$ for $\theta \in \mathbb{R}^p$.
2: **Constant step-size:** $\gamma > 0$
3: **Hyper-parameters:** $\sigma \in [0, 1]$, $\alpha \geq 0$, $\beta > \gamma$, $\epsilon = 10^{-8}$.
4: **Initialization:** time step $k \leftarrow 0$, parameter vector $\theta_0$, $v_0 = 0$.
5: **for** $k = 1$ **to** K **do**
6: $\quad g_k = \nabla\mathcal{J}(\boldsymbol{\theta}_k)$
7: $\quad \boldsymbol{v}_{k+1} \leftarrow \sigma\boldsymbol{v}_k + (1 - \sigma)\boldsymbol{g}_k^2$
8: $\quad \hat{\boldsymbol{v}}_{k+1} \leftarrow \boldsymbol{v}_{k+1}/(1 - \sigma^k)$
9: $\quad \boldsymbol{\theta}_{k+1} \leftarrow \boldsymbol{\theta}_k - \gamma_k\left(\boldsymbol{g}_k/(\sqrt{\hat{\boldsymbol{v}}_{k+1}} + \epsilon)\right)$
10: **return** $\theta_{K+1}$

---

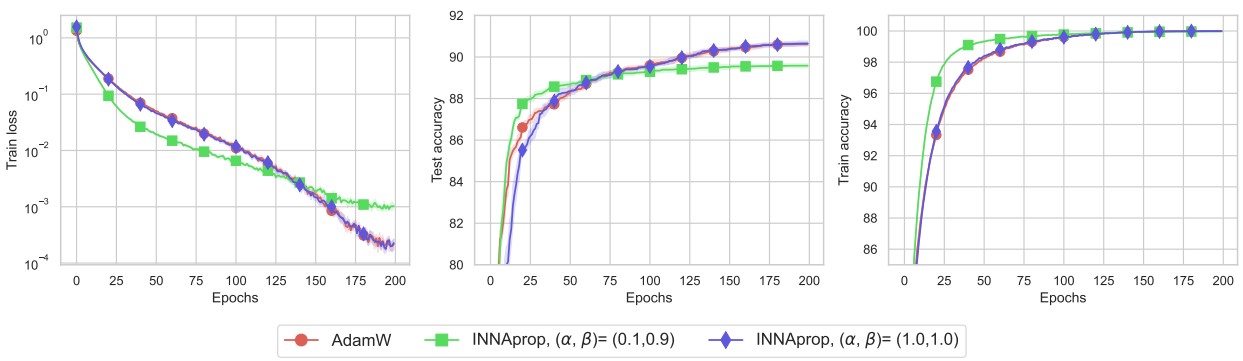

Figure 8: Training VGG11 on CIFAR10. Left: train loss, middle: test accuracy (%), right: train accuracy (%), with 8 random seeds. AdamW is considered without momentum to illustrate the connection with INNAprop, the two curves are indistinguishable.

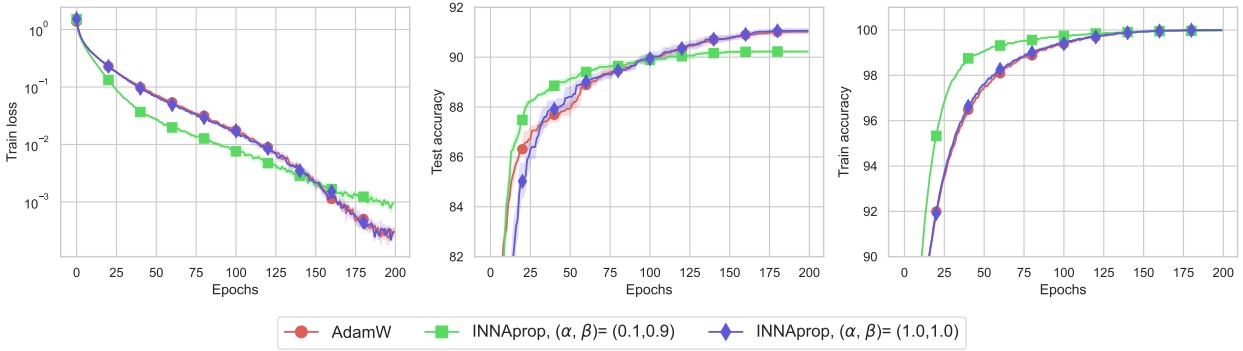

Figure 9: Training ResNet18 on CIFAR10. Left: train loss, middle: test accuracy (%), right: train accuracy (%), with 8 random seeds. AdamW is considered without momentum to illustrate the connection with INNAprop, the two curves are indistinguishable.

## C  Alternative discretizations

### C.1  A variant of INNAprop with momentum

**The algorithm.**  We follow the rationale behind the algorithm RMSprop with momentum (Graves, 2013). We therefore start with Equation (DIN) using the RMSprop proxy for the gradient as in (4), a direct discretization gives:

$$v_{k+1} = \sigma v_k + (1 - \sigma)g_k^2$$
$$\frac{\theta_{k+1} - 2\theta_k + \theta_{k-1}}{\gamma} + \alpha\frac{\theta_k - \theta_{k-1}}{\gamma} + \beta\frac{\frac{g_k}{\sqrt{v_{k+1}}+\epsilon} - \frac{g_{k-1}}{\sqrt{v_k}+\epsilon}}{\gamma} + \frac{g_{k-1}}{\sqrt{v_k}+\epsilon} = 0.$$

Rearranging terms, we have

$$v_{k+1} = \sigma v_k + (1 - \sigma)g_k^2$$
$$\theta_{k+1} = \theta_k + (1 - \alpha\gamma)(\theta_k - \theta_{k-1}) - \beta\gamma\left(\frac{g_k}{\sqrt{v_{k+1}}+\epsilon} - \frac{g_{k-1}}{\sqrt{v_k}+\epsilon}\right) - \gamma^2\frac{g_{k-1}}{\sqrt{v_k}+\epsilon}$$

Let us introduce a momentum variable $m_k = \theta_{k-1} - \theta_k$ to obtain:

$$v_{k+1} = \sigma v_k + (1 - \sigma)g_k^2 \tag{8}$$

$$m_{k+1} = (1 - \alpha\gamma)m_k + \gamma^2\frac{g_{k-1}}{\sqrt{v_k}+\epsilon} + \beta\gamma\left(\frac{g_k}{\sqrt{v_{k+1}}+\epsilon} - \frac{g_{k-1}}{\sqrt{v_k}+\epsilon}\right) \tag{9}$$

$$\theta_{k+1} = \theta_k - m_{k+1} \tag{10}$$

As previously we might have an equivalent rewrite of this recursion which allows for in place updates. For this we rewrite Equation (9) as

$$m_{k+1} = am_k + bg_k - cg_{k-1}. \tag{11}$$

where $a = (1 - \alpha\gamma)$, $b = \beta\gamma$ and $c = \gamma(\beta - \gamma)$. Writing $\tilde{m}_k = m_k - \frac{c}{a}g_{k-1}$, we have

$$\tilde{m}_{k+1} = m_{k+1} - \frac{c}{a}g_k$$
$$= am_k + bg_k - cg_{k-1} - \frac{c}{a}g_k$$
$$= a\left(m_k - \frac{c}{a}g_{k-1}\right) + \left(b - \frac{c}{a}\right)g_k$$
$$= a\tilde{m}_k + \left(b - \frac{c}{a}\right)g_k.$$

Therefore, using this identity, we may rewrite the following

$$m_{k+1} = am_k + bg_k - cg_{k-1},$$
$$\theta_{k+1} = \theta_k - m_{k+1}$$

as

$$\tilde{m}_{k+1} = a\tilde{m}_k + \left(b - \frac{c}{a}\right)g_k,$$
$$\theta_{k+1} = \theta_k - \tilde{m}_{k+1} - \frac{c}{a}g_k.$$

Recalling that $a = (1 - \alpha\gamma)$, $b = \beta\gamma$ and $c = \gamma(\beta - \gamma)$. Finally, we get the following recursion which is an alternative way to integrate RMSprop to INNA:

$$v_{k+1} = \sigma v_k + (1 - \sigma)g_k^2 \tag{12}$$

$$\tilde{m}_{k+1} = (1 - \alpha\gamma)\tilde{m}_k + \gamma^2\left(\frac{1 - \alpha\beta}{1 - \alpha\gamma}\right)\frac{g_k}{\sqrt{v_{k+1}} + \epsilon} \tag{13}$$

$$\theta_{k+1} = \theta_k - \tilde{m}_{k+1} - \frac{\gamma(\beta - \gamma)}{1 - \alpha\gamma}\frac{g_k}{\sqrt{v_{k+1}} + \epsilon} \tag{14}$$

but as shown below through numerical experiments the resulting algorithm performs poorly. Indeed, the factor $\gamma^2$ is poorly scaled for 32 bits or lower machine precision.

**Numerical experiments.** Using CIFAR-10 dataset, we train a VGG11 network with the momentum version of INNAprop with the hyperparameters $(\alpha, \beta) = (0.1, 0.9)$ above. We used a cosine annealing scheduler with $\gamma_0 = 10^{-3}$ and no weight decay. As seen in Figure 10, the training loss stops decreasing between the 125th and 150th epochs. Upon closely examining the algorithm in this regime, we observe that at the end of training, $\gamma_k^2$ falls below the numerical precision, resulting in unstable behavior in Equation (13).

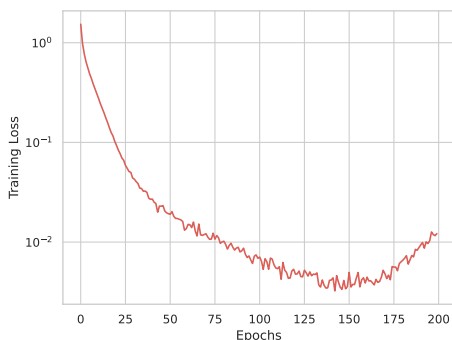

Figure 10: The version of INNA with momentum of Section C.1 is an unstable method.

## C.2 An approach à la Adam

In this section, we mimic the process for deriving Adam from the heavy ball with a RMSprop proxy, see, e.g., Kingma & Ba (2014); Ruder (2016), by simply replacing the heavy ball by DIN[6]. We call this optimizer DINAdam.

From (DIN), we infer the discretization:

$$\frac{\theta_{k+1} - 2\theta_k + \theta_{k-1}}{\gamma^2} + \alpha\frac{\theta_{k+1} - \theta_k}{\gamma} + \beta\frac{g_k - g_{k-1}}{\gamma} + g_k = 0. \tag{15}$$

---

[6]Note that DIN with $\beta = 0$ boils down to the heavy ball method.

Rearranging terms, we have

$$\theta_{k+1} = \theta_k - \frac{\gamma^2}{1+\alpha\gamma}g_k + \frac{1}{1+\alpha\gamma}(\theta_k - \theta_{k-1}) - \frac{\beta\gamma}{(1+\alpha\gamma)}(g_k - g_{k-1}) \tag{16}$$

By introducing the new variable $m_k = (\theta_{k-1} - \theta_k)/\eta$ and setting $\eta > 0$, we can rewrite equation (16) as:

$$m_{k+1} = \frac{1}{(1+\alpha\gamma)}m_k + \frac{\gamma^2}{(1+\alpha\gamma)\eta}g_k + \frac{\beta\gamma}{(1+\alpha\gamma)\eta}(g_k - g_{k-1}) \tag{17}$$

$$\theta_{k+1} = \theta_k - \eta m_{k+1} \tag{18}$$

To follow the Adam spirit, we set $\sigma_1 = \frac{1}{(1+\alpha\gamma)}$ and $(1-\sigma_1) = \frac{\gamma^2}{(1+\alpha\gamma)\eta}$. Solving for $\gamma$, we get

$$\frac{\alpha\gamma}{1+\alpha\gamma} = \frac{\gamma^2}{(1+\alpha\gamma)\eta} \Rightarrow \gamma = \frac{\eta}{\alpha}$$

Then, we find the following recursion:

$$m_{k+1} = \sigma_1 m_k + (1-\sigma_1)g_k + \beta\alpha\sigma_1(g_k - g_{k-1}) \tag{19}$$

$$\theta_{k+1} = \theta_k - \eta m_{k+1} \tag{20}$$

From Equation (19), we make a change of variable $\tilde{m}_k = m_k - \alpha\beta g_{k-1}$.

$$\tilde{m}_{k+1} = \sigma_1 \tilde{m}_k + (1 - \sigma_1 + \beta\alpha\sigma_1 - \beta\alpha)g_k \tag{21}$$

$$\theta_{k+1} = \theta_k - \eta(\tilde{m}_{k+1} + \alpha\beta g_k) \tag{22}$$

Using the usual RMSprop constants $\sigma_2 \in [0,1]$ and $\epsilon > 0$, we obtain:

$$v_{k+1} = \sigma_2 v_k + (1-\sigma_2)g_k^2 \tag{23}$$

$$\tilde{m}_{k+1} = \sigma_1 \tilde{m}_k + (1-\sigma_1)(1-\alpha\beta)g_k \tag{24}$$

$$\theta_{k+1} = \theta_k - \eta\frac{\tilde{m}_{k+1} + \alpha\beta g_k}{\sqrt{v_{k+1}} + \epsilon} \tag{25}$$

---

**Algorithm 4** DINAdam

---

1: **Objective function:** $\mathcal{J}(\theta)$ for $\theta \in \mathbb{R}^p$.
2: **Constant step-size:** $\gamma > 0$
3: **Hyper-parameters:** $(\sigma_1, \sigma_2) \in [0,1]^2$, $\alpha, \beta > 0$, $\epsilon = 10^{-8}$.
4: **Initialization:** $\theta_0$, $v_0 = 0$, $\tilde{m}_0 = 0$.
5: **for** $k = 1$ **to** K **do**
6: $\quad g_k = \nabla\mathcal{J}(\boldsymbol{\theta}_k)$
7: $\quad \boldsymbol{v}_{k+1} \leftarrow \sigma_2 \boldsymbol{v}_k + (1-\sigma_2)\boldsymbol{g}_k^2$
8: $\quad \tilde{\boldsymbol{m}}_{k+1} \leftarrow \sigma_1 \tilde{\boldsymbol{m}}_k + (1-\sigma_1)(1-\alpha\beta)\boldsymbol{g}_k$
9: $\quad \boldsymbol{\theta}_{k+1} \leftarrow \boldsymbol{\theta}_k - \gamma\frac{\tilde{\boldsymbol{m}}_{k+1}+\alpha\beta\boldsymbol{g}_k}{\sqrt{\boldsymbol{v}_{k+1}}+\epsilon}$
10: **return** $\theta_{K+1}$

---

**Remark 5** The way RMSprop is integrated within INNAprop and DINAdam is different. In INNAprop, RMSprop is incorporated directly during the discretization process of Equation (4) and in particular the RMSprop scaling apply to all gradient terms in the recursion. However, in DINAdam, RMSprop is added only at the last step, as shown in Equation (23), and only affects the momentum term. This is how RMSprop was combined with heavy ball to obtain Adam.

**Remark 6** After setting $\alpha = 1$ and $\beta = 0$, we obtain Adam update rules. If $\beta \neq 0$, DINAdam is qualitatively very close to the NAdam algorithm, with a different choice of hyperparameters. This is seen for from Algorithm 4, to be compared with Table 5 (for $\alpha\beta < 1$). In both cases, the update is a weighted average between the current gradient and a discounted average of past gradients, making it possible to recover each recursion equivalently with different hyperparameter choices (including the step-size).

## D    Scheduler procedures

**Cosine annealing (Loshchilov & Hutter, 2016).** Let $\gamma_k$ represent the learning rate at iteration $k$, $T_{\max}$ be the maximum number of iterations (or epochs), and $\gamma_{\min}$ be the minimum learning rate (default value is 0). The learning rate $\gamma_k$ at iteration $k$ is given by:

$$\gamma_k = \gamma_{\min} + \frac{1}{2}(\gamma_0 - \gamma_{\min})\left(1 + \cos\left(\frac{k}{T_{\max}}\pi\right)\right)$$

This scheduler was employed in all image classification experiments except for ViT.

**Cosine annealing with linear warmup (Radford et al., 2018).** Let $\gamma_k$ represent the learning rate at iteration $k$, $\gamma_{\min}$ the minimum learning rate, $\gamma_0$ the initial learning rate, $T_{\text{warmup}}$ the number of iterations for the warmup phase, and $T_{\text{decay}}$ the iteration number after which the learning rate decays to $\gamma_{\min}$. The learning rate is defined as follows:

$$\gamma_k = \begin{cases} \gamma_0 \cdot \frac{k}{T_{\text{warmup}}}, & \text{if } k < T_{\text{warmup}} \\ \gamma_{\min} + \frac{1}{2}\left(\gamma_0 - \gamma_{\min}\right)\left(1 + \cos\left(\pi \cdot \frac{k - T_{\text{warmup}}}{T_{\text{decay}} - T_{\text{warmup}}}\right)\right), & \text{if } T_{\text{warmup}} \leq k \leq T_{\text{decay}} \\ \gamma_{\min}, & \text{if } k > T_{\text{decay}} \end{cases}$$

This scheduler was applied in experiments involving training GPT-2 from scratch and for ViT.

**Linear schedule with linear warmup (Hu et al., 2021).** Let $\gamma_k$ represent the learning rate at iteration $k$ and $T_{\max}$ be the maximum number of iterations, $T_{\text{warmup}}$ be the number of warmup steps, and $\gamma_{\min}$ be the minimum learning rate after warmup (default value is typically set to the initial learning rate, $\gamma_0$). The learning rate $\gamma_k$ at iteration $k$ is given by:

$$\gamma_k = \begin{cases} \gamma_0 \cdot \frac{k}{T_{\text{warmup}}} & \text{if } k < T_{\text{warmup}}, \\ \gamma_0 \cdot \left(1 - \frac{k - T_{\text{warmup}}}{T_{\max} - T_{\text{warmup}}}\right) & \text{otherwise.} \end{cases}$$

This scheduler was used for fine-tuning GPT-2 with LoRA.

## E    Choosing hyperparameters $\alpha$ and $\beta$ for INNAprop

### E.1    Comparison with AdamW

For VGG and ResNet training on CIFAR10, the literature suggest using initial learning rate $\gamma_0 = 10^{-3}$ with a learning rate schedule (Mishchenko & Defazio, 2023; Defazio & Mishchenko, 2023; Yao et al., 2021; Zhuang et al., 2020). Our experiment fix a cosine scheduler where $T_{\max} = 200$ and $\gamma_{\min} = 0$ as it achieves a strong baseline for AdamW (Loshchilov & Hutter, 2016; Mishchenko & Defazio, 2023). We set weight decay $\lambda = 0.1$. Then, we tune the initial learning rate $\gamma_0$ among $\{10^{-4}, 5 \times 10^{-4}, 10^{-3}, 5 \times 10^{-3}, 10^{-2}\}$. In Figure 11, we report the performance in terms of training loss and test accuracy for AdamW. These results confirm the usage of $\gamma_0 = 10^{-3}$.

(a) Performance rankings with VGG11.

| $\gamma_0$ | Train loss | Test accuracy (%) |
|---|---|---|
| $10^{-3}$ | 0.00041 | 91.02 |
| $5 \times 10^{-3}$ | 0.00047 | 90.86 |
| $5 \times 10^{-4}$ | 0.00048 | 90.79 |
| $10^{-2}$ | 0.00057 | 90.41 |
| $10^{-4}$ | 0.00081 | 88.49 |

(b) Performance rankings with ResNet18.

| $\gamma_0$ | Train loss | Test accuracy (%) |
|---|---|---|
| $10^{-3}$ | 0.00040 | 92.1 |
| $5 \times 10^{-3}$ | 0.00049 | 91.84 |
| $5 \times 10^{-4}$ | 0.00094 | 92.32 |
| $10^{-2}$ | 0.00057 | 90.41 |
| $10^{-4}$ | 0.0018 | 87.85 |

Figure 11: Comparative performance of the training loss and test accuracy according to $\gamma_0$. We trained VGG11 and ResNet18 models on CIFAR10 for 200 epochs.

# F   Additional experiments

## F.1   CIFAR10 experiments

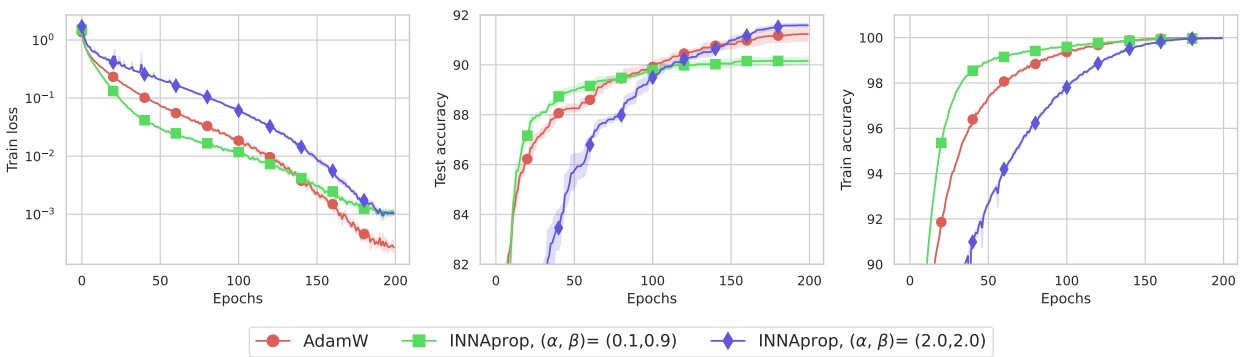

Figure 12: Training ResNet18 on CIFAR10. Left: train loss, middle: test accuracy (%), right: train accuracy (%), with 8 random seeds.

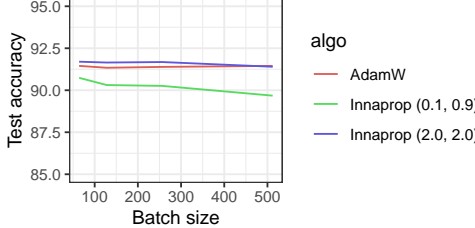

Figure 13: Influence of the minibatch size for training ResNet18 on CIFAR10. The learning rate is tuned for each batch size. The influence of the batch size is relatively moderate for all three algorithms.

### F.2 Food101 experiments

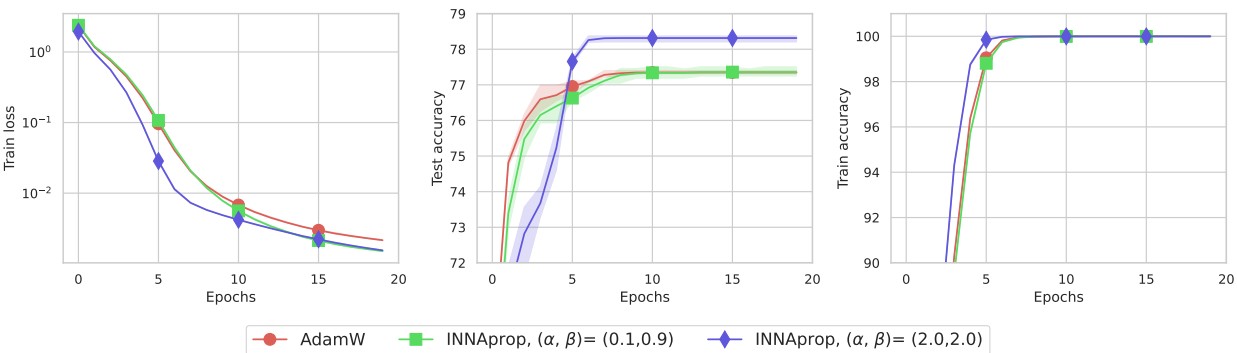

Figure 14: Finetuning a ResNet18 on Food101, same as Figure 4 for ResNet18. Left: train loss, middle: test accuracy (%), right: train accuracy (%), with 3 random seeds.

### F.3 ImageNet

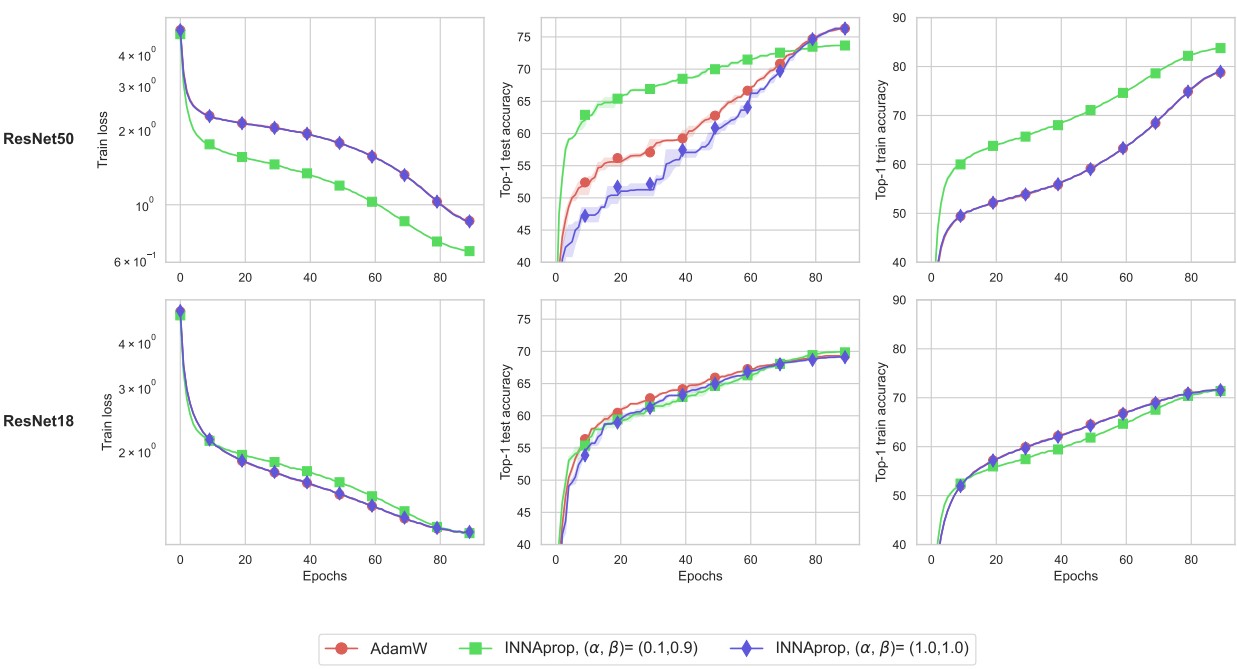

Figure 15: Training a ResNet50 (top) and ResNet18 (bottom) on ImageNet. Left: train loss, middle: Top-1 test accuracy (%), right: Top-1 train accuracy (%). 3 random seeds.

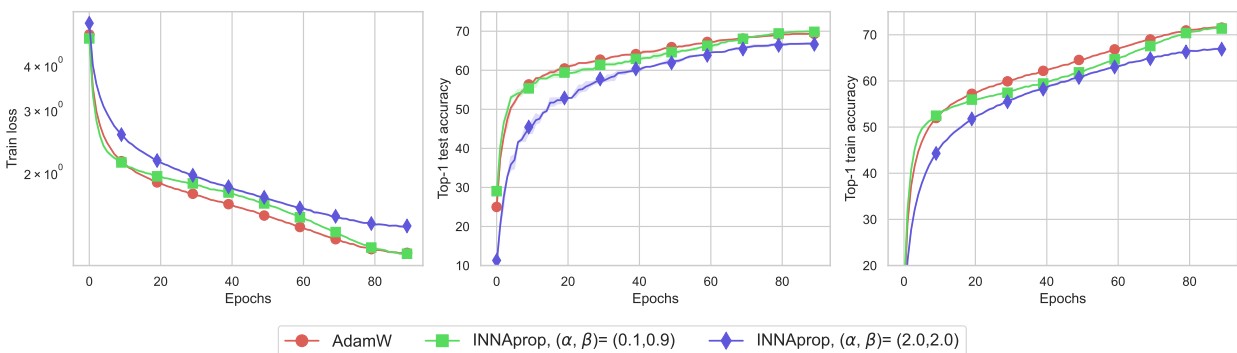

Figure 16: Training ResNet18 on ImageNet. Left: train loss, middle: test accuracy (%), right: train accuracy (%), with 3 random seeds. The superiority of INNAprop is consistent from epochs 70 to 90. Note that we chose 90 epochs to be consistent with the hyper-parameter settings reported in the literature Chen et al. (2023b).

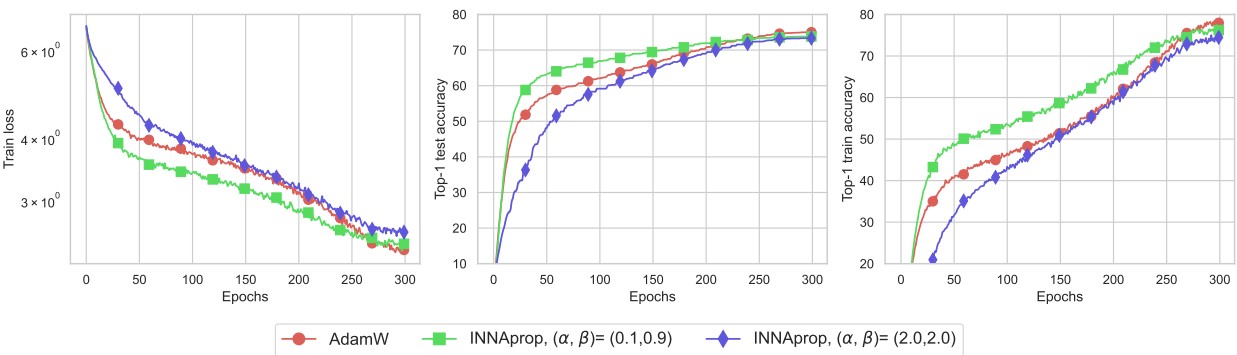

Figure 17: Fast training ViT/B-32 on ImageNet with weight decay $\lambda = 0.01$ for INNAprop $(\alpha, \beta) = (0.1, 0.9)$. Left: train loss, middle: test accuracy (%), right: train accuracy (%), with 3 random seeds.

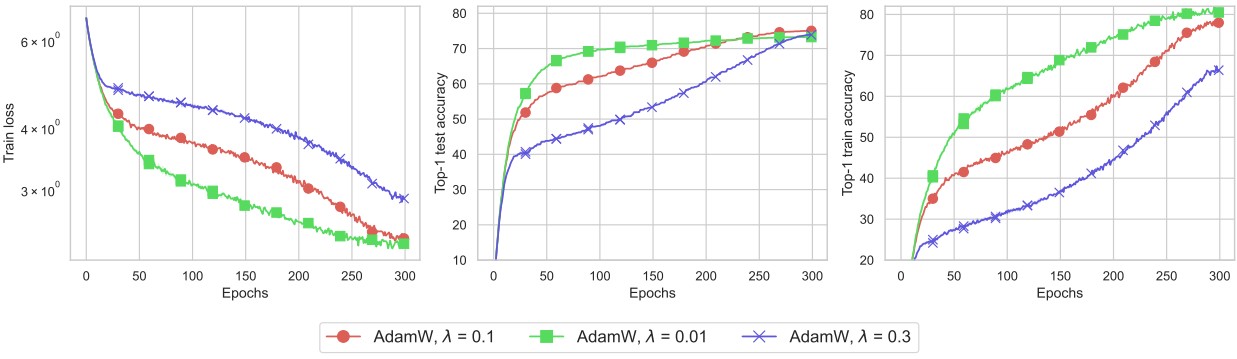

Figure 18: Tuning the weight decay for the AdamW optimizer on the ViT/B-32 ImageNet training, with 3 random seeds.

## F.4    Heatmap for preliminary tuning of $\alpha$ and $\beta$

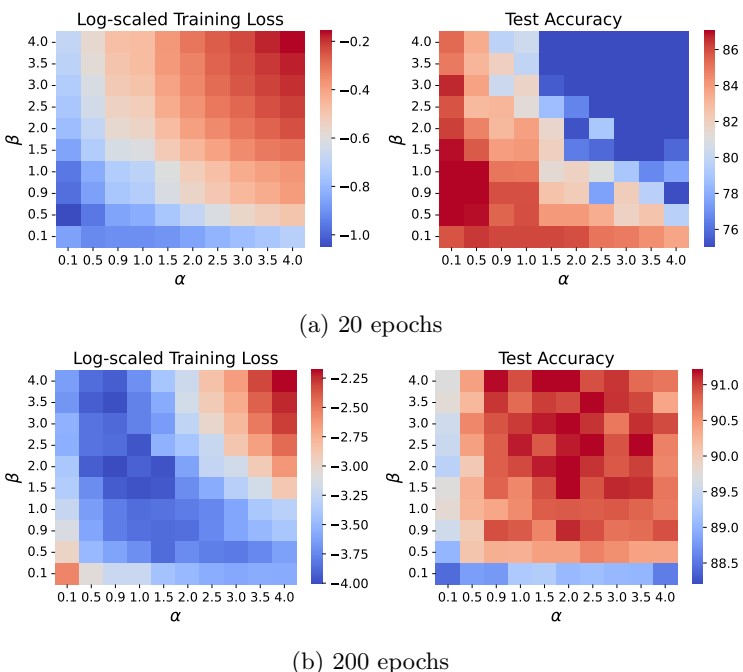

Figure 19: Log-scale training loss and test accuracies for $(\alpha, \beta)$ hyperparameters with VGG11 on CIFAR10 at different epochs. Optimal learning rate $\gamma_0 = 10^{-3}$, weight decay $\lambda = 0$.

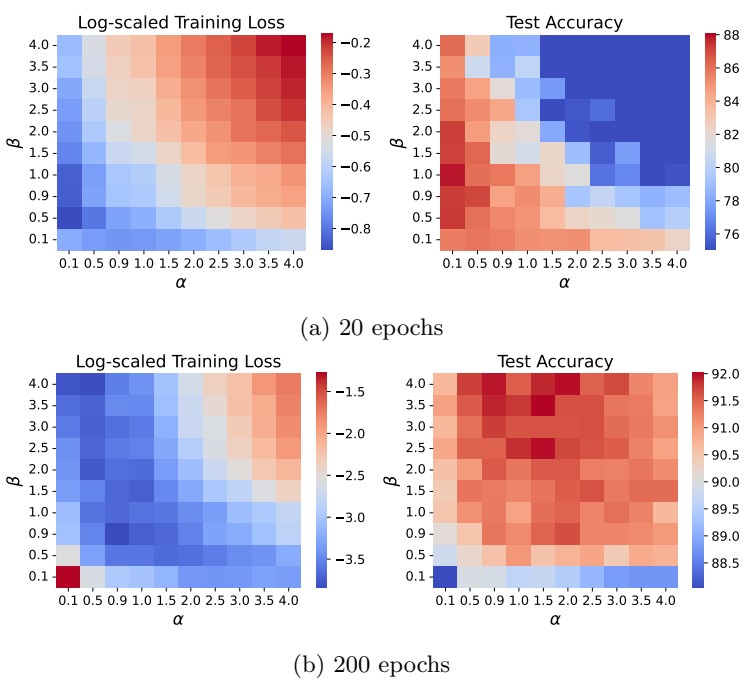

Figure 20: Log-scale training loss and test accuracies for $(\alpha, \beta)$ hyperparameters with ResNet18 on CIFAR10 at different epochs. Optimal learning rate $\gamma_0 = 10^{-3}$, weight decay $\lambda = 0.01$.

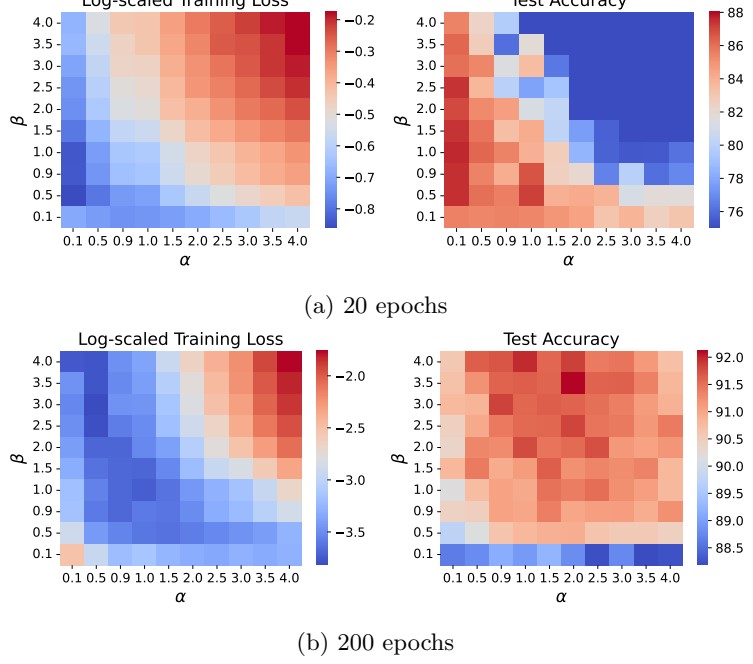

Figure 21: Log-scale training loss and test accuracies for $(\alpha, \beta)$ hyperparameters with ResNet18 on CIFAR10 at different epochs. Optimal learning rate $\gamma_0 = 10^{-3}$, weight decay $\lambda = 0$.

# G  Experimental Setup

## G.1  CIFAR-10

We used custom training code based on the PyTorch tutorial code for this problem. Following standard data-augmentation practices, we applied random horizontal flips and random offset cropping down to 32x32, using reflection padding of 4 pixels. Input pixel data was normalized by centering around 0.5.

| Hyper-parameter | Value |
|---|---|
| Architecture | VGG11 and ResNet18 |
| Epochs | 200 |
| GPUs | 1×V100 |
| Batch size per GPU | 256 |
| Baseline LR | 0.001 |
| Seeds | 8 runs |

| Hyper-parameter | Value |
|---|---|
| Baseline schedule | cosine |
| Weight decay $\lambda$ | 0.01 |
| $\beta_1, \beta_2$ (for AdamW) | 0.9, 0.999 |
| $\sigma$ (for INNAprop) | 0.999 |

## G.2  Food101

We used the pre-trained models available on PyTorch for VGG11 and ResNet18.[7]

| Hyper-parameter | Value |
|---|---|
| Architecture | VGG11 and ResNet18 |
| Epochs | 200 |
| GPUs | 1×V100 |
| Batch size per GPU | 256 |
| Baseline LR | 0.001 |
| Seeds | 3 runs |

| Hyper-parameter | Value |
|---|---|
| Baseline schedule | cosine |
| Weight decay $\lambda$ | 0.01 |
| $\beta_1, \beta_2$ (for AdamW) | 0.9, 0.999 |
| $\sigma$ (for INNAprop) | 0.999 |

---

[7]https://pytorch.org/vision/stable/models.html

### G.3 ImageNet

We used the same code-base as for our CIFAR-10 experiments, and applied the same preprocessing procedure. The data-augmentations consisted of PyTorch's RandomResizedCrop, cropping to 224x224 followed by random horizontal flips. Test images used a fixed resize to 256x256 followed by a center crop to 224x224.

#### G.3.1 ResNet18

| Hyper-parameter | Value |
|---|---|
| Architecture | ResNet18 |
| Epochs | 90 |
| GPUs | 4×V100 |
| Batch size per GPU | 64 |
| Baseline LR | 0.001 |
| Seeds | 3 runs |

| Hyper-parameter | Value |
|---|---|
| Baseline schedule | cosine |
| Weight decay $\lambda$ | 0.01 |
| $\beta_1, \beta_2$ (for AdamW) | 0.9, 0.999 |
| $\sigma$ (for INNAprop) | 0.999 |

#### G.3.2 ResNet50

| Hyper-parameter | Value |
|---|---|
| Architecture | ResNet18 |
| Epochs | 90 |
| GPUs | 4×V100 |
| Batch size per GPU | 64 |
| Baseline LR | 0.001 |
| Mixed precision | True |
| Seeds | 3 runs |

| Hyper-parameter | Value |
|---|---|
| Baseline schedule | cosine |
| Weight decay $\lambda$ | 0.1 |
| $\beta_1, \beta_2$ (for AdamW) | 0.9, 0.999 |
| $\sigma$ (for INNAprop) | 0.999 |

#### G.3.3 ViT/B-32

| Hyper-parameter | Value |
|---|---|
| Architecture | ViT/B-32 |
| Epochs | 300 |
| GPUs | 8×A100 |
| Batch size per GPU | 128 |
| Baseline LR | 0.001 |
| Seeds | 5000 |

| Hyper-parameter | Value |
|---|---|
| Baseline schedule | cosine |
| Warmup | linear for 30 epochs |
| Weight decay $\lambda$ | 0.1 |
| $\beta_1, \beta_2$ (for AdamW) | 0.9, 0.999 |
| $\sigma$ (for INNAprop) | 0.999 |

### G.4 GPT2 from scratch

We followed the NanoGPT codebase [8] and we refer to (Brown et al., 2020) as closely as possible, matching the default batch-size and schedule.

| Hyper-parameter | Value |
|---|---|
| Architecture | GPT-2 |
| Batch size per gpu | 12 |
| Max Iters | 100000 |
| GPUs | 4×A100 |
| Dropout | 0.0 |
| Baseline LR | refer to (Brown et al., 2020) |
| Warmup Steps | 500 |

| Hyper-parameter | Value |
|---|---|
| Seeds | 5000 |
| Weight decay $\lambda$ | 0.1 |
| $\beta_1, \beta_2$ (for AdamW) | 0.9, 0.95 |
| $\sigma$ (for INNAprop) | 0.99 |
| Gradient Clipping | 1.0 |
| Float16 | True |

---

[8] https://github.com/karpathy/nanoGPT

### G.5 GPT-2 with LoRA

We followed the LoRA codebase [9] and we refer to (Hu et al., 2021) as closely as possible, matching the default batch-size, training length, and schedule. We train all of our GPT-2 models using AdamW (Loshchilov & Hutter, 2017) and INNAprop on E2E dataset Novikova et al. (2017) with a linear learning rate schedule for 5 epochs. We report the mean result over 3 random seeds; the result for each run is taken from the best epoch.

| Hyper-parameter | Value |
|---|---|
| Architecture | GPT-2 |
| Batch size per gpu | 8 |
| Epochs | 5 |
| GPUs | 1×A100 |
| Dropout | 0.1 |
| Baseline LR | 0.0002 |
| Warmup steps | 500 |

| Hyper-parameter | Value |
|---|---|
| Seeds | 3 runs |
| Weight decay $\lambda$ | 0.01 |
| $\beta_1, \beta_2$ (for AdamW) | 0.9, 0.98 |
| $\sigma$ (for INNAprop) | 0.98 |
| Learning Rate Schedule | Linear |
| LoRA $\alpha$ | 32 |

---

[9] https://github.com/microsoft/LoRA

