

Figure 1: Addition of the Muon optimizer to the paper experiment. We used the Modded-NanoGPT repository codebase, which is part of a collaborative competition for fast GPT-2 training on a 8xH100 cluster. For Muon, we took the minibatch size, schedule and weight decay from this repository. We adapted the number of GPUs to fit our cluster, also impacting the total number of training tokens. We also used the same nanoGPT architecture as we used for AdamW and INNAprop and the OpenWebText dataset.

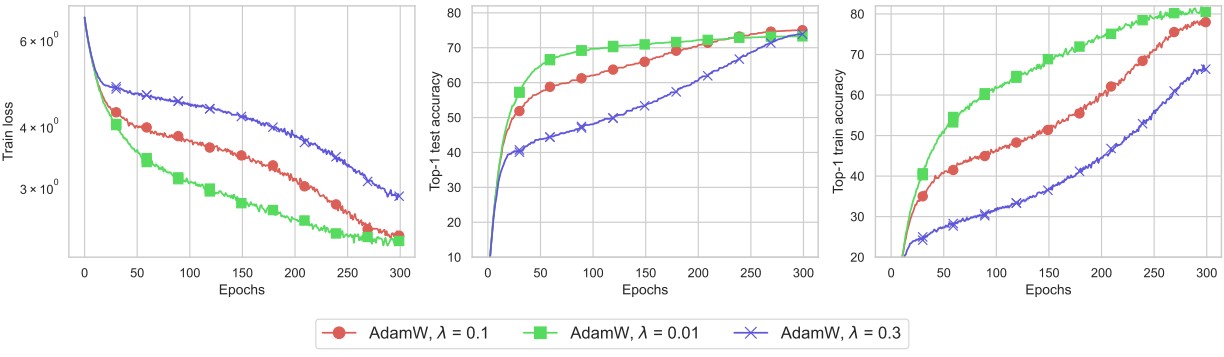

Figure 2: Variation of the weight decay for AdamW for training the vision transformer. This aligns with empirical claims in the literature that a weight decay of 0.1 is optimal for AdamW in this setting, and that the reported performance of INNA-prop is not due to any hidden weight decay parameter.