# OpenReview forum: "A second-order-like optimizer with adaptive gradient scaling for deep learning"
_TMLR — Accepted by TMLR_

### Review · Reviewer_Coz7 · 2025-07-11

**Summary Of Contributions:**

The paper proposes a new optimizer which combines the core concepts of INNA and RMSProp/Adagrad style preconditioning of the gradient.
An extensive experimental evaluation the performance of the optimizer in comparison to AdamW on machine learning tasks is provided.

**Audience:**

Yes

**Broader Impact Concerns:**

None.

**Claims And Evidence:**

Yes

**Requested Changes:**

This section is more "Questions":

* You mention other recent optimizers (ScheduleFree, Muon etc.). It would be interesting to see one of these methods for the GPT-2 tasks in comparison, where INNAprop performs very well. For me it would be sufficient to reuse the reported hyperparameter values of these methods, as the setup should be very similar and one can assume that the reported values are tuned. (If INNAprop performs worse than those methods, this would not alter my rating of the submission.)

* Rather unrelated to the proposed method: in Fig 5 AdamW seems to be better than Sophia, however the original Sophia paper claims a big advantage over AdamW. Can you comment on this phenomenon? Is it because you were not able to reproduce the (supposedly) superior performance of Sophia, or because you tuned AdamW better than the baseline from the Sophia paper?

* From the Resnet50/Imagenet plot, it seems that green performs very well in the beginning, but only blue is able to match the final acccuracy of AdamW. Have you tried to schedule $\alpha$ and $\beta$ over training in order to get fast initial convergence, and good final performance?

Minor:

* is the term "geometrical intelligence" standardly used? I am not sure what it means.

**Strengths And Weaknesses:**

Strengths:

* The algorithm design is very clearly motivated and easy to follow: it combines mechanisms from two directions (INNA and RMSProp) with the hope of getting the best of the two worlds.
* The experimental evaluation is very fair and thorough work. It covers several data domains and model architectures, runs multiple seeds, and properly accounts for hyperparameter tuning.

Weaknesses:

* The main question I had after reading the paper is: what exactly is the motivation to propose yet another optimization method? As mentioned above, the steps of the derivation are clear, and the experiments also are insightful. From the current version, it seems that the method lacks either a theoretical access or a clear empirical advantage over AdamW.

* The method seems to be quite sensitive to the choice of $\alpha$ and $\beta$.

---

> ### Author Response · Authors · 2025-08-28
> **Response to reviewer Coz7**
>
> >-  The main question I had after reading the paper is: what exactly is the motivation to propose yet another optimization method? As mentioned above, the steps of the derivation are clear, and the experiments also are insightful. From the current version, it seems that the method lacks either a theoretical access or a clear empirical advantage over AdamW.
>
> We believe that there is still room to improve deep network training and that the ideas behind the DIN continuous dynamical system constitutes a relevant direction. While we could not show a systematic clear advantage over AdamW, we show that the method  performs reasonably well on a diversity of tasks and sometimes significantly better than AdamW with hyperparemeters tuned for AdamW and not INNAProp.
>
> >- The method seems to be quite sensitive to the choice of $\alpha$ and $\beta$.
>
> The reviewer is right. The choice of hyperparameters matters a lot. This is the reason why we conducted an extensive preliminary study. This is also typical for concurent optimizers such as AdamW.
>
> >1. You mention other recent optimizers (ScheduleFree, Muon etc.). It would be interesting to see one of these methods for the GPT-2 tasks in comparison, where INNAprop performs very well. For me it would be sufficient to reuse the reported hyperparameter values of these methods, as the setup should be very similar and one can assume that the reported values are tuned. (If INNAprop performs worse than those methods, this would not alter my rating of the submission.)
>
> We thank the reviewer for this suggestion. We conducted a new experiment using the Muon optimizer on GPT2 small (see supplement pdf attached). INNAprop ranks first and Muon falls in between INNAprop and AdamW in terms of test error, consistently along iterations. We used the Modded-NanoGPT repository codebase,  which is part of a collaborative competition for fast GPT-2/Freeweb training on a 8xH100 cluster. For Muon, we took the minibatch size, schedule and weight decay from this repository (once more hyper-optimization is on the side of concurrent solvers, here Muon). We adapted the number of GPUs to fit our cluster, also impacting the total number of training tokens. We also used the same nanoGPT architecture as we used for AdamW and INNAprop and the OpenWebText dataset. We will add this experiment in the revision. The structure of Muon, AdamW and INNAprop are very similar. The schedule-free optimizer has a different structure (no schedule) and we could not reproduce the GPT2small experiment in a direct way. Note that contrary to Muon/GPT2 small speedrun, the codebase for ScheduleFree/GPT2 small is not available and reproducing it would require significantly more work, trial/errors.
>
>
> >2. Rather unrelated to the proposed method: in Fig 5 AdamW seems to be better than Sophia, however the original Sophia paper claims a big advantage over AdamW. Can you comment on this phenomenon? Is it because you were not able to reproduce the (supposedly) superior performance of Sophia, or because you tuned AdamW better than the baseline from the Sophia paper?
>
> Thank you for the question. Despite careful training/coding, we were not able to reproduce the superior performance of Sophia reported in the original paper. This seems to be a known reproducibility issue in the community — similar discrepancies have been documented by other users (see [GitHub issue #46](https://github.com/Liuhong99/Sophia/issues/46)).
>
> >3. From the Resnet50/Imagenet plot, it seems that green performs very well in the beginning, but only blue is able to match the final accuracy of AdamW. Have you tried to schedule and over training in order to get fast initial convergence, and good final performance?
>
> Thank you for the suggestion. We did experiment with scheduling $\alpha$ and $\beta$ during training, aiming to combine fast initial convergence with strong final performance. However, in our preliminary studies, we were not able to find a stable schedule that consistently improved both phases — most attempts either harmed convergence or degraded generalization. But this remains an open direction, and we believe more principled or adaptive scheduling strategies must definitely be explored in future work.
>
> >* is the term "geometrical intelligence" standardly used? I am not sure what it means.
>
> What we meant by "geometrical intelligence" — a phrase we can reformulate — is that our dynamical system leverages rich geometric information on the geometry of the loss, Hessians, past gradients, derivatives of past gradients, momentum. Empirically, this combination contributes to globally better performance.

---

> > ### Comment · Reviewer_Coz7 · 2025-09-03
> > **Response to revision**
> >
> > Dear authors,
> >
> > thank you for the clear and detailed responses, and for the additional experiments. I have no further questions, and will recommend this paper for publication.

---

### Review · Reviewer_MYuc · 2025-07-18

**Summary Of Contributions:**

This paper empirically investigates an optimizer which combines the Inertial Newton Algorithm (INNA, [Castera et al., JMLR 2021](https://www.jmlr.org/papers/volume22/19-1024/19-1024.pdf))) with RMSProp. The Inertial Newton Algorithm can be summarized as below:

$$\theta_{t+1}=\theta_t + \gamma_t((\frac{1}{\beta}-\alpha)\theta_t-\frac{1}{\beta}\psi_t - \beta g_t)$$

$$\psi_{t+1}=\psi_t + \gamma_t((\frac{1}{\beta}-\alpha)\theta_t-\frac{1}{\beta}\psi_t)$$

where $t$ is the step, $\theta_t$ the model parameter, $g_t$ the gradient, and $\gamma_t$ the learning-rate; $\psi_t$ is a slot variable which tracks a decayed running average of the model parameter, and $\alpha$, $\beta$ are hyper-parameters.

Intuitively, INNA is considered as a variant of SGD with heavy-ball momentum and weight-decay. More precisely, in the above, $-\alpha\theta_t$ which comes from the "friction" term in INNA's framework, is corresponding to weight-decay; and the $\frac{1}{\beta}(\theta_t-\psi_t)$ term can be considered as some kind of momentum, because it calculates the difference between the current model parameter and a running average of previous ones, and uses it to achieve an "inertia" effect in the update. INNA is heuristically derived from a 2nd order ODE as an analogue to Newtonian Mechanics.

Based on INNA, this paper proposes INNAProp, by replacing the $g_t$ term with a RMSProp-style normalized gradient:

$$\theta_{t+1}=\theta_t + \gamma_t((\frac{1}{\beta}-\alpha)\theta_t-\frac{1}{\beta}\psi_t - \beta \frac{g_t}{\sqrt{\hat{\nu}_{t+1}}+\epsilon})$$

$$\psi_{t+1}=\psi_t + \gamma_t((\frac{1}{\beta}-\alpha)\theta_t-\frac{1}{\beta}\psi_t)$$

where $\hat{\nu}_{t}$ is the bias-corrected running average of gradient squares, as usual. So INNAProp can be viewed as a variant of AdamW, of which the (first) momentum and (decoupled) weight-decay are implemented in INNA's style.

In addition, this paper proposes to rearrange the formula above, by updating $\psi_{t+1}$ before $\theta_{t+1}$, and using $\psi_{t+1}$ instead of $\psi_{t}$ when updating $\theta_{t+1}$. As below:

$$\psi_{t+1}=\psi_t + \gamma_t((\frac{1}{\beta}-\alpha)\theta_t-\frac{1}{\beta}\psi_t)$$

$$\theta_{t+1}=\theta_t + \gamma_t(\frac{1-\alpha\beta}{\beta-\gamma_t}\theta_t-\frac{1}{\beta-\gamma_t}\psi_{t+1} - \beta \frac{g_t}{\sqrt{\hat{\nu}_{t+1}}+\epsilon})$$

Although this rearrangement is mathematically equivalent, the authors claim that "it can save the memory footprint of the algorithm by avoiding the storage of $\psi_t$". However, I don't believe this rearrangement can achieve any observable effect in saving the memory footprint in practice. (Because, what this rearrangement saves is the temporary scratch space of one tensor; during update, the computer is going to calculate the updates for a lot of tensors, while the temporary scratch space is reused. So the effect of this rearrangement is negligible. -- If the authors have different opinions, please show me an ablation experiment comparing the memory usage and running speed.)

**Audience:**

Yes

**Broader Impact Concerns:**

None.

**Claims And Evidence:**

No

**Requested Changes:**

Besides revision of the theoretical writing and more convincing experimental results, as described above, I'm also curious about how INNAProp performs with the [original Transformer on WMT](https://arxiv.org/abs/1706.03762). The original Transformer, especially with PostNorm, usually requires momentum and learning-rate warmup to be trained properly; it should be good to know if the INNA-style momentum can achieve the same on the Transformer-WMT task. A good repo where one can find the SOTA Transformer-WMT settings is [init2winit](https://github.com/google/init2winit).

**Strengths And Weaknesses:**

## Strengths:

I think the proposal of INNAProp is quite interesting. There is no issue with the direction of this study; the optimizer is well-motivated. And I think the experimental results are positive, in the sense that INNAProp passes the "sanity check" -- I believe it can work, and indeed it seems to work -- which is not trivial.

I don't believe INNAProp has any theoretical advantage over AdamW; and I don't believe that, with thorough hyper-parameter tuning, one can consistently outperform another in practice. But what makes INNAProp unique, is that the slot variable $\psi_t$ only keeps track of the model parameter, but does not depend on the gradient at all; yet it can achieve an effect of momentum. This might be useful in some scenarios, for example when the gradient calculation is unreliable.

## Weakness:

In the technical details, this paper needs to improve a lot.

1. The theoretical part is difficult to follow; especially Section 2.2, _Derivation of the algorithm_: I couldn't understand how the algorithm was derived from the 2nd order ODE, at all; the narration refers to Equation (7) and (8), but Equation (8) is not presented in the main paper -- it jumps to the Appendix. And I still couldn't quite get the essence after reading the Appendix, until I read Castera et al.'s JMLR paper. You are not making a big theoretical advancement in this paper, so you have a very good existing framework in Castera et al.'s paper to follow -- yet why is the writing of this paper so messy? This is almost irritating. Furthermore, the "rewriting the algorithm in another coordinate system to save memory footprint" part -- as I discussed above, this doesn't make much sense. Don't mention it. Save some space.

2. One big flaw in the experiments, is that the "$\alpha$" hyper-parameter in INNAProp is almost the same as the $\lambda$ hyper-parameter for weight-decay. So the settings in this paper, which align $\lambda$ for INNAProp and AdamW, but tune $\alpha$ separately, is not actually a "head-to-head" comparison between INNAProp and AdamW. This discrepancy seems to manifest in the ImageNet experiments in Section 3.2: the one performing better on training set performs worse on test set, and vice versa -- we know that ImageNet data is noisy and it requires very strong weight decay to achieve the best test accuracy -- so these results probably just reflect the difference of weight-decay strength; they don't indicate either INNAProp or AdamW is better.

3. More generally, on ImageNet with ResNet and ViT, there are certain numbers which we refer to as SOTA -- one has to achieve those numbers in order to make the results convincing. It's unlikely that INNAProp can achieve a better number than the best tuned AdamW on these tasks, but there is a chance that INNAProp can have a traning loss curve that shows faster convergence. The [LION paper](https://arxiv.org/abs/2302.06675) and its repo can be a good place to find the SOTA settings. For Transformer LMs, one needs to carefully determine the model size and training steps, and do thorough hyper-parameter tuning to make the results convincing -- the default settings in most open-sourced repos are far from optimal. The [Chinchilla paper](https://arxiv.org/abs/2203.15556) is the scheme to follow for such evaluation. I understand this is not easy; it requires a lot of experiments to write a convincing paper on a new optimizer. But this is the way it is.

---

> ### Author Response · Authors · 2025-08-28
> **Response to reviewer Myuc (1/2)**
>
> > 1. The theoretical part is difficult to follow; especially Section 2.2, Derivation of the algorithm: I couldn't understand how the algorithm was derived from the 2nd order ODE, at all; the narration refers to Equation (7) and (8), but Equation (8) is not presented in the main paper -- it jumps to the Appendix.
>
> There was a labeling issue in the previous version — Equation (8) was mistakenly placed only in the Appendix. We are sorry about this confusion which will be corrected.
>
>
> > 2. And I still couldn't quite get the essence after reading the Appendix, until I read Castera et al.'s JMLR paper. You are not making a big theoretical advancement in this paper, so you have a very good existing framework in Castera et al.'s paper to follow -- yet why is the writing of this paper so messy? This is almost irritating.
>
>
> Following the reviewers' comments and also the comments of 3aip, we will modify the presentation. We will start with INNA and remark that the continuous time change of variable works with a general vector field. This will considerably shorten the presentation. As in the general comments section, we emphasize that this is a shortcut and the original derivation in Appendix B of the discretization allows for  verifying that this indeed constitutes a discretization of the continuous system  involving second order features in equation (5). This will be presented in a remark and developed in the Appendix.
>
> > 3. Furthermore, the "rewriting the algorithm in another coordinate system to save memory footprint" part -- as I discussed above, this doesn't make much sense. Don't mention it. Save some space.
>
> According to Hugging Face’s [Ultrascale Playbook](https://huggingface.co/spaces/nanotron/ultrascale-playbook?section=high-level_overview),
> the main bottleneck in training large-scale models is *memory usage*. Optimizer states (e.g., momentum, denominator for Adam, the added variable in INNA) can indeed consume a significant portion of the memory required for training.
>
> For example, for LLaMA 8B, optimizer states alone consume 75 GB out of the total 160 GB required for a full optimizer step, while standard GPUs provide only 80 GB of memory.
>
>  Training often use implementation tricks such as mixed precision (https://arxiv.org/pdf/2110.02861) in order to cope with optimizer states. Saving an optimizer state has a crucial impact on the scale of models which can be trained using a given computing architecture.
>
> The INNA version of the algorithm (end of page 16) requires storing both $\psi_k$ and $\psi_{k+1}$, which prohibits *in-place computation* for $\psi$ and effectively adds one optimizer state.
>
> Our proposed modification allows in-place computation for $\psi$. We agree that the original presentation was unclear; we will rephrase and polish this important step, emphasizing that it is an equivalent form of the update that saves one optimizer state by enabling in-place computation for $\psi$.
>
>
>
> > 4. One big flaw in the experiments, is that the "$\alpha$" hyper-parameter in INNAProp is almost the same as the hyper-parameter for weight-decay. So the settings in this paper, which align for INNAProp and AdamW, but tune separately, is not actually a "head-to-head" comparison between INNAProp and AdamW. This discrepancy seems to manifest in the ImageNet experiments in Section 3.2: the one performing better on training set performs worse on test set, and vice versa -- we know that ImageNet data is noisy and it requires very strong weight decay to achieve the best test accuracy -- so these results probably just reflect the difference of weight-decay strength; they don't indicate either INNAProp or AdamW is better.
>
> We respectfully disagree, while understanding that the role of $\alpha$ may be seen as a variation of uncoupled weight decay. However, their nature is different: weight decay is a *regularization term* à la Tikhonov, anchoring weights toward low norms, Whereas $\alpha$ is a *friction parameter* that plays a central role in the theory of accelerated methods independently of regularization (a long bibliography — starting with the pioneering work of Su, Boyd, and Candès, *A differential equation for modeling Nesterov’s accelerated gradient method: Theory and insights* — attests to this fact).
>
>  In order to provide direct evidence in the empirical spirit we chose, we conducted additional experiments on the vision transformer model and ImageNet dataset: AdamW with weight decay $\lambda = 0.01, 0.1, 0.3$. We found that $\lambda = 0.1$, the value reported by Defazio and Mischenko (2023) which we used, provides the best results in terms of test accuracy (see supplement pdf attached). We hope that this experiment adresses the concern of the reviewer on this matter. This experiment and a summary of the reviewer's comment will be reported in the revised version of the manuscript.

---

> ### Author Response · Authors · 2025-08-28
> **Response to reviewer Myuc (2/2)**
>
> > 5. More generally, on ImageNet with ResNet and ViT, there are certain numbers which we refer to as SOTA -- one has to achieve those numbers in order to make the results convincing. It's unlikely that INNAProp can achieve a better number than the best tuned AdamW on these tasks, but there is a chance that INNAProp can have a training loss curve that shows faster convergence. The LION paper and its repo can be a good place to find the SOTA settings. For Transformer LMs, one needs to carefully determine the model size and training steps, and do thorough hyper-parameter tuning to make the results convincing -- the default settings in most open-sourced repos are far from optimal. The Chinchilla paper is the scheme to follow for such evaluation. I understand this is not easy; it requires a lot of experiments to write a convincing paper on a new optimizer. But this is the way it is.
>
> For Imagenet, we actually have used parameters from the LION paper for AdamW. For example, we do 90 epochs with minibatch 256 as in Chen et al (2023), the LION paper, which mentions "the optimal batch size for AdamW is 256" in the comment on Figure 8. We also use similar schedules etc ... This is explicit in Section 3.2 and we believe that this adresses the first part of the reviewer's comment.
> Regarding the Chinchilla paper, our focus is different. We aim to compare optimizers on fixed architectures and to show that INNAprop is a robust, general-purpose optimizer across diverse settings. By contrast, Chinchilla explores the effect of carefully hyperoptimizing all parameters (including those of LLMs) to maximize efficiency in a  specific regimes. Our contribution is not to re-tune  detail for LLM training, but to provide an optimizer that is (hopefully) broadly applicable.
>
> However, for comparison with existing state of the art on LLM training, we added an experiment with the Muon optimizer (see supplement pdf attached). We took the code and hyperparameters from the Moded-Nanogpt codebase, which is part of a collaborative competition for fast GPT-2 training on a 8xH100 cluster. See the response to reviewer Coz7 first requested change for more details on this matter. INNAprop ranks first followed by MUON and AdamW. This will be included in the revision.
>
>
> > 6. Besides revision of the theoretical writing and more convincing experimental results, as described above, I'm also curious about how INNAProp performs with the original Transformer on WMT. The original Transformer, especially with PostNorm, usually requires momentum and learning-rate warmup to be trained properly; it should be good to know if the INNA-style momentum can achieve the same on the Transformer-WMT task. A good repo where one can find the SOTA Transformer-WMT settings is init2winit.
>
> We thank the reviewer for this suggestion. The proposed repository is coded using Jax and we conducted all our experiments using Pytorch...  Setting up this experiment on our GPU cluster requires a significant amount of additional resources and we could not unfortunately conduct it during the time allowed for response.

---

> ### Comment · Reviewer_MYuc · 2025-08-28
> **Thanks for the response**
>
> ## Regarding the memory footprint,
>
> I think the authors did not get my point (which is also raised by Reviewer 3aip) yet:
>
> The point is, even without the "in-place computation" trick, simply a naive implementation of the update rule
>
> $$\theta_{t+1}=\theta_t + \gamma_t((\frac{1}{\beta}-\alpha)\theta_t-\frac{1}{\beta}\psi_t - \beta \frac{g_t}{\sqrt{\hat{\nu}_{t+1}}+\epsilon})$$
>
> $$\psi_{t+1}=\psi_t + \gamma_t((\frac{1}{\beta}-\alpha)\theta_t-\frac{1}{\beta}\psi_t)$$
>
> would **NOT** require additional storage of $\psi_t$. This is because the compiler (assume there is a compiler) will do things like the following:
>
> $\psi_t$ consists of several per-layer tensors, denoted by $\psi_t^{(1)}$, ..., $\psi_t^{(n)}$.
>
> The computer will process the tensors one by one, say from $\psi_t^{(1)}$ to $\psi_t^{(n)}$.
>
> For tensor $\psi_t^{(i)}$, it will use the first formula above to calculate $\theta_{t+1}^{(i)}$; then keeping $\psi_t^{(i)}$, use the second formula to calculate $\psi_{t+1}^{(i)}$. At this point, $\psi_t^{(i)}$ will not be used further, so its memory will be released.
>
> Therefore, the extra memory usage caused by $\psi_t^{(i)}$ is only temporary during the calculation of $\psi_{t+1}^{(i)}$. For large models, there are a lot of layers; while the results $\psi_{t+1}^{(i)}$ for all $i=1,\ldots,n$ should be kept for later usage, the memory for $\psi_t^{(i)}$ is reused at each layer. Compared to the total memory consumed by all $\psi_{t+1}^{(i)}$ ($i=1,\ldots,n$), a temporary one tensor of one layer is negligible. This is what I mean by "temporary scratch space".
>
> Moreover, the compiler will actually divide each tensor further into smaller blocks, and it will probably fuse the calculation of $\theta_{t+1}^{(i)}$ and $\psi_{t+1}^{(i)}$ together. As a result, the temporary scratch space required is the size of the smaller blocks. It will **NOT** cause any significant memory usage.
>
> In short, the compiler will be smart enough to eliminate the unnecessary storage of large tensors.
>
> Well, at least this is what will happen in JAX; if the authors are using an eager mode of PyTorch which does not have a compiler, and it is observed that $\psi_t$ is kept until the calculation of $\psi_{t+1}$ for all layers is complete -- this is more of an issue for PyTorch.
>
> ## For the role of the $\alpha$ hyper-parameter,
>
> It is certainly debatable, that it originates from the Newton friction term theoretically does not mean it won't interact or entangle with the weight-decay term. Indeed, at least for $\theta_{t+1}$, the two terms have the same effect. Therefore, when comparing INNAProp with AdamW, the hyper-parameters $(\alpha,\lambda)$ should be considered as a pair together, and it's not directly comparable with the $\lambda$ hyper-parameter of AdamW. One has to grid-search all of them in order to do a fair comparison.

---

> > ### Author Response · Authors · 2025-09-01
> > **Second response to reviewers comments, thanks for your feedback**
> >
> > **About the memory footprint.**
> >
> > Thank you for your detailed comments. We will simplify the presentation as follows:
> > - Start with INNA and plugin the RMS-prop term in the main text as suggested by the referee (version 0 of the algorithm).
> > - Keep the iterates in Algorithm 1 unchanged (version 1 of the algorithm).
> > - Add a remark saying that Algorithm 1 presents an update equivalent to version 0, which allows a naive implementation of the algorithm with "in-place" update for psi. This remark will also mention that a clever implementation of version 0, or additional computational layers such as compilers, could handle this type of optimization automatically.
> >
> > This will be short. We thank you for your very interesting comments on code compilation.
> >
> >
> > **About the role of alpha.**
> >
> > We optimized alpha and beta by grid search in preliminary CIFAR-10 experiments. For larger experiments, we compare performance with fixed lambda (optimized for Adam-W according to the literature) and choose alpha from the preliminary experiment. There is almost no ($\alpha$, $\lambda$) tuning for INNAprop and tuning them would only get better results. This protocol was adopted to avoid criticisms about over-optimizing hyperparameters for the proposed method. This protocol is in favor of AdamW. We will add a remark about the fact that optimizing hyperparameters could increase INNAprop performances and insist on the relevance of weight decay, as proposed by the reviewer.
> >
> > That said, we apologize, but the $\alpha$ parameter has a different effect compared to weight decay. This is seen by considering $u = \theta - \phi$, our method computes
> > $$u_{t+1} = u_t - \gamma_t \beta g_t,$$
> > which is independent of $\alpha$. On the other hand decoupled weight decay only modifies the $\theta$ equation, yielding
> > $$u_{t+1} = u_t - \gamma_t(\beta g_t + \lambda \theta_t),$$
> > which clearly depend on $\lambda$, while the corresponding $\psi$ update remains independent of $\lambda$.
> > In other words, $\alpha$ and $\lambda$ affect different coordinates in the $(\theta, \psi)$ space. So they are not interchangeable and the parameter $\alpha$ cannot be subsumed into weight decay, even if the equation for $\theta$ shows superficially similar linear terms. We will add a remark about this connection in the text and leave further exploration of the connection between $\alpha$ and weight decay for future work.

---

> > > ### Comment · Reviewer_MYuc · 2025-09-13
> > > **Thanks for the discussion**
> > >
> > > **Some further comments regarding the role of alpha:**
> > >
> > > > There is almost no ($\alpha$, $\lambda$) tuning for INNAprop and tuning them would only get better results
> > >
> > > Yes, I believe so; please do the experiments and show the better results. Because currently, (1) INNAprop is not really outperforming AdamW in all settings; and (2) AdamW may not be tuned to the SOTA performance.
> > >
> > > > This protocol was adopted to avoid criticisms about over-optimizing hyperparameters for the proposed method
> > >
> > > There will always be such criticisms; I believe the only way is to beat the SOTA numbers. I'm leaning towards requiring more experiments for this work.
> > >
> > > > So they are not interchangeable and the parameter $\alpha$ cannot be subsumed into weight decay, even if the equation for $\theta$ shows superficially similar linear terms. We will add a remark about this connection in the text and leave further exploration of the connection between $\alpha$ and weight decay for future work.
> > >
> > > I'm not saying $\alpha$ can be subsumed into weight decay; I get the math. I'm saying that $\alpha$ and weight-decay are likely having similar effects on $\theta$, so separating them and claiming that the comparison is under the same settings and the "protocol is in favor of AdamW" is questionable. Please do not leave further exploration of the connection between $\alpha$ and weight decay for future work. Please do it in this work.

---

> > > > ### Author Response · Authors · 2025-09-18
> > > > **Thanks for the discussion**
> > > >
> > > > *Yes, I believe so; please do the experiments and show the better results. Because currently, (1) INNAprop is not really outperforming AdamW in all settings; and (2) AdamW may not be tuned to the SOTA performance.*
> > > >
> > > > May we emphasize that we do not claim INNAprop to outperform AdamW across all settings. If any of our phrasing inadvertently suggested otherwise, we are glad to clarify and adjust it. Our intention has been to highlight quite constant wins rather than universal superiority, and we fully recognize that AdamW is likely to remain the more suitable choice for certain classes of problems.
> > > >
> > > > *There will always be such criticisms; I believe the only way is to beat the SOTA numbers. I'm leaning towards requiring more experiments for this work.*
> > > >
> > > > We will make clear in our revision that AdamW has been extensively tuned by a large research community, with ongoing fine-grained refinements that can be sensitive to small changes in computing environments. Our method is therefore evaluated against this long history of careful tuning; when it performs well without additional hyperparameter tuning, we find this especially reassuring. We believe, or at least hope, that these principles are consistent with TMLR policy.
> > > >
> > > >
> > > >
> > > >
> > > >
> > > > *I'm not saying
> > > >  can be subsumed into weight decay; I get the math. I'm saying that
> > > >  and weight-decay are likely having similar effects on
> > > > , so separating them and claiming that the comparison is under the same settings and the "protocol is in favor of AdamW" is questionable. Please do not leave further exploration of the connection between
> > > >  and weight decay for future work. Please do it in this work.*
> > > >
> > > > We are truly sorry, but we do not believe they have similar effects except in exceptional cases (critical in the mathematical case). We already provided the mathematical justification in the last round and supporting experiments in the one before, and at this stage, we do not know what to do save to reiterate that these points will be detailed in the final version of our work.

---

### Review · Reviewer_3aip · 2025-08-14

**Summary Of Contributions:**

- This paper introduces INNAprop, a deep learning optimizer derived by combining the INNA (dynamical inertial Newton) method with RMSprop’s adaptive gradient scaling approach. The resulting optimizer is designed to enjoy second-order-information-like benefits, while maintaining the same memory requirements as AdamW, making it suitable for deep learning.
- The authors show through extensive experiments that INNAprop is competitive to AdamW across a variety of benchmarks, including image classification datasets with modern CNN and ViT architectures, as well as language modeling and fine-tuning tasks with GPT-2.

**Audience:**

Yes

**Broader Impact Concerns:**

No concern.

**Claims And Evidence:**

Yes

**Requested Changes:**

## Critical comments to address

1. It seems that the starting of the derivations of INNAprop with momentum in Appendix C.2 is exactly the same as the starting of the derivations of INNAprop in Appendix B. How do they result in different methods as stated in Remark 1 (b)?
2. Perhaps I missed something. The authors claim to "improve the memory footprint of INNA by saving a memory slot." However, examining INNA's update rules in Table 5, while INNA maintains vector $\psi_k$, it doesn't compute $v_k$ (unlike INNAprop). Shouldn't their memory footprints stay the same ?
3. **Unclear claims in Remark 2**: Several statements require clarification, such as "When $\beta = 0$, we recover a modified version of RMSprop with momentum" (discussed in Appendix B.1 for Adam, not RMSprop) and "If β ≠ 0, DINAdam is very close to NAdam algorithm" (lacking further discussion despite Table 5 providing update rules for known optimizers).
4. The authors claim that INNAprop benefits from second-order information. I have concern about this claim. First, I'm not familiar with the literature of dynamical inertial Newton and finite difference approximation applied on it. In my opinion, there is a distinction between using actual second-order information and its approximations. How effective is the finite difference approximation applied to dynamical inertial Newton methods? Given the extensive literature cited, can the authors provide additional evidence for this research direction's general effectiveness?
5. There is this statement "by setting $\alpha = \beta = 1$, we empirically recover the behavior of AdamW" in Remark 2. Isn't this claim addressed in Table 3 for INNAprop ($\alpha = \beta = 1$) ? I find the result of INNAprop ($\alpha = \beta = 1$) in Table 3 strange, as it is not discussed elsewhere in the paper.
6. Regarding Figure 3, the authors claim that "in the ImageNet experiments, we evaluated INNAprop for rapid early training and optimal final test accuracy without tuning $(\gamma_0, \alpha, \beta)$. For ViT-B/32 with $\lambda = 0.1$, INNAprop achieved lower training loss and higher final test accuracy than AdamW (75.23 vs. 75.02)." However, for ViT-B/32, Figure 3 shows AdamW achieving lower training loss with similar final test accuracy to INNAprop. Thus, the claimed advantage is not clearly evident.
7. Regarding Figure 14, the authors claim that "our algorithm with $(\alpha, \beta) = (0.1, 0.9)$ outperforms AdamW in test accuracy (70.12 vs 69.34)." While INNAprop slightly outperforms AdamW in final test accuracy (70.12 vs. 69.34), it performs worse during most of training. This raises concerns about whether training was deliberately stopped at an advantageous moment for INNAprop. For instance, what will happen if we leave the training longer ?
8. Based on comments 6 and 7, the claim that the algorithm "either matches or outperforms AdamW" on large-scale vision models appears unsupported.

## Suggestions for strengthening the work

- Add missing citations for OpenWebText and E2E datasets.
- Include proper discussion of DenseNet, which currently appears only in the abstract and Table 2.
- Add DINAdam and INNAprop to Table 5 for clearer comparison.
- **Include INNA comparisons in the main paper**: While I acknowledge Appendix F.5 presents INNA experiments in language modeling, these results should be appeared in the main text. Since INNAprop already demonstrates strong language modeling performance, it would be more valuable to evaluate whether INNAprop outperforms INNA in computer vision tasks with a clear margin. Such comparisons would better justify the proposed method's advantages over its direct predecessor.
- As mentioned previously, explore INNAprop's robustness through batch size variation experiments to showcase additional advantages of second-order information beyond faster convergence.

**Strengths And Weaknesses:**

## Strengths

- **Rigorous algorithmic foundation**: The algorithmic derivation is well-grounded in continuous-time ODE theory with detailed step-by-step discretization. The explicit variable usage ensures memory equivalence with AdamW (see Algorithm 1 and derivation in Section 2). The connection to previous inertia-based and adaptive methods is appropriately covered.
- **Fair experimental protocol**: The hyperparameter tuning protocol is transparently described and consistently favors AdamW. The authors make considerable effort to ensure AdamW is used in its optimized or literature-preferred settings, reducing confounding effects from suboptimal baselines (as detailed in Table 1 and corresponding sections).
- **The empirical evaluation is relatively comprehensive in scope**: Experiments are reported across vision (CIFAR-10, Food101 and ImageNet with different model architectures ResNet, VGG, DenseNet, ViT) and language modeling (GPT-2, including both pretraining and LoRA fine-tuning). The evaluation spans both small- and large-scale scenarios, including training speed and accuracy comparisons.
- **Strong language modeling performance**: The empirical results for GPT-2 pretraining and fine-tuning demonstrate that INNAprop outperforms AdamW, representing a notable achievement for the proposed method.

## Weaknesses
- **Overreliance on “favoring AdamW” protocol**: While the decision to always optimize/tune AdamW hyperparameters is commendable for fairness, it sometimes leads to the impression that INNAprop’s absolute best performance is unexplored, potentially underrepresenting its peak capability. I'm not asking the authors to do massive hyperparameters tuning, but some interesting questions might be ignored. For instance, given its purported second-order-like information benefits, how does INNAprop respond to batch size variations? Such experiments could illuminate its robustness characteristics.
- **Insufficient intuitive explanation**: Although the derivation is mathematically clear, the paper lacks intuitive discussion about why INNAprop should outperform alternatives like DINAdam or INNA. While INNAprop incorporates second-order information for RMSprop scaling, which is an advantage compared to RMSprop, the specific advantages over INNA remain unclear. This gap makes the motivation for developing INNAprop appear incomplete, merely another deep learning optimizer with integrated second-order-like information rather than a principled advancement.
- **Unclear method relationships**: The connections and comparisons between INNAprop, DINAdam, INNA, NAdam, and AdamW lack clarity (see critical comments 1-4 below).
- **Questionable vision results**: The empirical results in computer vision are not entirely convincing (see critical comments 6-8 below).

---

> ### Author Response · Authors · 2025-08-28
> **Response to reviewer 3aip (1/3)**
>
> > **Overreliance on “favoring AdamW” protocol**: While the decision to always optimize/tune AdamW hyperparameters is commendable for fairness, it sometimes leads to the impression that INNAprop’s absolute best performance is unexplored, potentially underrepresenting its peak capability. I'm not asking the authors to do massive hyperparameters tuning, but some interesting questions might be ignored. For instance, given its purported second-order-like information benefits, how does INNAprop respond to batch size variations? Such experiments could illuminate its robustness characteristics.
>
> We did not have time to conduct the proposed experiment with a variation of the batch size. We plan to conduct and include such an experiment on a relatively small network similarly as Figure 2 in the SCION paper (Training Deep Learning Models with Norm-Constrained LMOs)
>
>
> > **Insufficient intuitive explanation:** Although the derivation is mathematically clear, the paper lacks intuitive discussion about why INNAprop should outperform alternatives like DINAdam or INNA. While INNAprop incorporates second-order information for RMSprop scaling, which is an advantage compared to RMSprop, the specific advantages over INNA remain unclear. This gap makes the motivation for developing INNAprop appear incomplete, merely another deep learning optimizer with integrated second-order-like information rather than a principled advancement.
>
> The referee is right. As explained in the general section, we will rework the presentation of INNAprop, see responses below and response to MYuc's second comment.
>
>
> > **Unclear method relationships:** The connections and comparisons between INNAprop, DINAdam, INNA, NAdam, and AdamW lack clarity (see critical comments 1-4 below).
>
> Again you are right, this part will be reworked to address the reviewers concerns (see below).
>
> > ** Questionable vision results:** The empirical results in computer vision are not entirely convincing (see critical comments 6-8 below).
>
> This part will be corrected also (see below)
>
>
> **Critical comments to address**
>
> > 1. It seems that the starting of the derivations of INNAprop with momentum in Appendix C.2 is exactly the same as the starting of the derivations of INNAprop in Appendix B. How do they result in different methods as stated in Remark 1 (b)?
>
> Following your comment and the comments of MYuc we will simplify the derivation, start with INNA and remove the presentation at the beginning of Appendix B. We emphasize that this is a shortcut and the original derivation in Appendix B of the discretization allows to verify that this heuristic derivation indeed constitutes a discretization of the continuous system involving second order features in (5). This will be presented in a remark.
>
>
>
> > 2. Perhaps I missed something. The authors claim to "improve the memory footprint of INNA by saving a memory slot." However, examining INNA's update rules in Table 5, while INNA maintains vector $\psi_k$, it doesn't compute
> $v_k$ (unlike INNAprop). Shouldn't their memory footprints stay the same ?
>
> The INNA version of the algorithm (at the end of page 16) requires to store both $\psi_k$ and $\psi_{k+1}$, prohibiting in place computation for $\psi$ and effectively adding one optimizer state. The modification that we propose allows to perform in place computation for $\psi$. We agree that the presentation was unclear, we will rephrase and polish the presentation of this step, insist that it is an equivalent form of the update which allows to save one optimizer state by allowing in place computation for the variable $\psi$.
>
>
> > 3. Unclear claims in Remark 2: Several statements require clarification, such as "When
> $\beta=0$, we recover a modified version of RMSprop with momentum" (discussed in Appendix B.1 for Adam, not RMSprop) and "If $\beta\neq0$, DINAdam is very close to NAdam algorithm" (lacking further discussion despite Table 5 providing update rules for known optimizers).
>
> We agree and will clarify these points as suggested. The purpose was to highlight connections with existing algorithms.

---

> ### Author Response · Authors · 2025-08-28
> **Response to reviewer 3aip (2/3)**
>
> > 4. The authors claim that INNAprop benefits from second-order information. I have concern about this claim. First, I'm not familiar with the literature of dynamical inertial Newton and finite difference approximation applied on it. In my opinion, there is a distinction between using actual second-order information and its approximations. How effective is the finite difference approximation applied to dynamical inertial Newton methods? Given the extensive literature cited, can the authors provide additional evidence for this research direction's general effectiveness?
>
> INNAprop continuous dynamics is first order in space and second order in time. However it seeks second order space information by looking a time variations of the vector field along optimization trajectories, which involves its Jacobian. This discussion takes place in continuous time and the main argument for this feature to transfer to the discrete algorithm is that for small step sizes, the continuous and discrete trajectories are close. We agree that INNAprop is a first-order (gradient) method in the sense that it does not use second-order derivatives explicitly (although the continuous formulation displays the term $\nabla^2 J \nabla J$). The claim that it leverages second-order information should be understood as a qualitative, heuristic argument, here in the context of deep network training. We will clarify this discussion in the revision.
>
>
>
> > 5. There is this statement "by setting $\alpha = \beta = 1$ , we empirically recover the behavior of AdamW" in Remark 2. Isn't this claim addressed in Table 3 for INNAprop ($\alpha = \beta = 1$
> ) ? I find the result of INNAprop ($\alpha = \beta = 1$) in Table 3 strange, as it is not discussed elsewhere in the paper.
>
> The reviewer is right. Remark 2 says that we obtain AdamW without momentum ($\beta_1 = 0$). In table 3, AdamW is used with the momentum parameters reported for this experimental setting. This experiment highlights that the momentum parameter is not well tuned in reported literature for AdamW in this experimental setting (we could set it to 0). This explains the differences in Table 3. We will remove the sentence "we empirically recover the behavior of AdamW" which is very misleading.
>
>
> > 6. Regarding Figure 3, the authors claim that "in the ImageNet experiments, we evaluated INNAprop for rapid early training and optimal final test accuracy without tuning $(\gamma_0,\alpha,\beta)$. For ViT-B/32 with $\lambda = 0.1$, INNAprop achieved lower training loss and higher final test accuracy than AdamW (75.23 vs. 75.02)." However, for ViT-B/32, Figure 3 shows AdamW achieving lower training loss with similar final test accuracy to INNAprop. Thus, the claimed advantage is not clearly evident.
>
>
> Yes, indeed. We kindly recall, however, that our method is not hypertuned, unlike AdamW, which has benefited from years of intensive optimization by the global research and engineering community. AdamW is the default optimizer in most deep learning frameworks (e.g., PyTorch, Hugging Face Transformers) and has been tuned across countless benchmarks and production pipelines. We agree that this particular experiment does not show a clear advantage for INNAprop. Our main point is that INNAprop remains competitive across a diversity of scenarios, despite the absence of comparable large-scale community-driven tuning.
>
>
> > 7.Regarding Figure 14, the authors claim that "our algorithm with
>  outperforms AdamW in test accuracy (70.12 vs 69.34)." While INNAprop slightly outperforms AdamW in final test accuracy (70.12 vs. 69.34), it performs worse during most of training. This raises concerns about whether training was deliberately stopped at an advantageous moment for INNAprop. For instance, what will happen if we leave the training longer ?
>
> INNAprop outperforms AdamW consistently from epoch 70 to 90. We trained for 90 epochs *as it is standard in the literature* (notably in the LION paper, mentioned by Reviewer MYuc). This stopping point is a choice exclusively  motivated by established practice rather than by selecting a favorable snapshot. Extending training beyond 90 epochs would be somewhat unnatural given the usual protocol.
>
>
> > 8.Based on comments 6 and 7, the claim that the algorithm "either matches or outperforms AdamW" on large-scale vision models appears unsupported.
>
> Yes, we will soften this claim: "offer close performances to AdamW and sometimes significantly better, notably in our LLM training experiments"

---

> ### Author Response · Authors · 2025-08-28
> **Response to reviewer 3aip (3/3)**
>
> **Suggestions for strengthening the work**
>
> 1. Add missing citations for OpenWebText and E2E datasets.
> 2. Include a proper discussion of DenseNet, which currently appears only in the abstract and Table 2.
> 3. Add DINAdam and INNAprop to Table 5 for clearer comparison.
> All these will be addressed in the revision.
>
> > 4. Include INNA comparisons in the main paper: While I acknowledge Appendix F.5 presents INNA experiments in language modeling, these results should be appeared in the main text. Since INNAprop already demonstrates strong language modeling performance, it would be more valuable to evaluate whether INNAprop outperforms INNA in computer vision tasks with a clear margin. Such comparisons would better justify the proposed method's advantages over its direct predecessor.
>
> We are grateful for these suggestions and will follow them in our revision.
>
> > 5. As mentioned previously, explore INNAprop's robustness through batch size variation experiments to showcase additional advantages of second-order information beyond faster convergence.
>
> As already mentioned, we did not have time to conduct this experiment but plan to include it in a revised version using relatively small networks, similarly as Figure 2 in the SCION paper (Training Deep Learning Models with Norm-Constrained LMOs).

---

> ### Comment · Reviewer_3aip · 2025-09-15
> **Thanks for the response**
>
> Thanks for the discussion and additional experiments. I've also read the other reviewers' comments. Overall, I see two main aspects for improvement: (1) the presentation and derivation of INNAProp, which has been sufficiently addressed in the rebuttal, including memory footprint, and (2) the experiments and discussion of results. I'm globally satisfied with the responses and additional experiments. I also appreciate that the authors plan to include experiments on batch size effects (even if not provided during rebuttal), which will strengthen the paper.
>
> I have one concern about the new experiments. The authors claim that "$\lambda = 0.1$, the value reported by Defazio and Mishchenko (2023) which we used, provides the best results in terms of test accuracy." However, Figure 2 suggests otherwise. The weight decay 0.01 shows significantly better training loss and accuracy, outperforming other values during early training and most of the training process. The $\lambda = 0.1$ only slightly outperforms $\lambda = 0.01$ for test accuracy at the end of training. Moreover, the claim that the weight decay 0.1 is optimal in the literature is questionable. While Defazio & Mishchenko (2023) use weight decay 0.1 for ViT (Table 13), Mishchenko & Defazio (2024) find that weight decay 0.05 significantly improve ViT training performance compared to weight decay 0.1.
>
> Finally, I respectfully disagree with Reviewer MYuc. I think proposing a new optimizer doesn't require beating SOTA, but rather demonstrating its promise and potential—which this work does. Given AdamW's extensive tuning over decades, the authors' choice to compare against it is reasonable.
>
> Despite my question for the new experiments (not sure if the authors can still reply, as the rebuttal period is closed), I recommend this paper for publication.
>
> [1] Aaron Defazio and Konstantin Mishchenko. Learning-Rate-Free Learning by D-Adaptation, ICML 2023.
>
> [2] Konstantin Mishchenko and Aaron Defazio. Prodigy: An Expeditiously Adaptive Parameter-Free Learner, ICML 2024.

---

> > ### Author Response · Authors · 2025-09-22
> > **Thanks a lot for your input**
> >
> > We are grateful for these insights and suggestions.
> >
> > On weight decay, we will moderate indeed our description and add an experiment with weight decay 0.05.
> >
> > For mini-batches, we will also vary the batch size for a moderately sized network.
> >
> > We are indebted to the referee for this fruitful exchange.

---

### Author Response · Authors · 2025-08-28
**General comments**

We thank all the referees for their thoughtful comments. Detailed responses are given after each report. We would like to single out three points:

- **Exposition.** Some redactional and explanatory aspects could indeed be improved or were not sufficiently pedagogical; we will revise to avoid repetitions and provide more direct motivations.

- **Derivation.** Following referees advices, for the derivation of INNAProp, we will proceed in a direct way: start from INNA and replace the gradient by the RMSProp direction. We nevertheless wish to emphasize that the model DIN, beyond its mechanical interpretation, can be viewed as a *superposition* of three dynamical systems: the gradient flow, the heavy-ball dynamics, and the Newtonian dynamics, i.e.,
$$  a \tfrac{d}{dt} \nabla J(\theta) + b \nabla J(\theta) = 0, \quad a,b>0.$$
  A straightforward computation shows that this system directly implies exponential convergence of the gradient to zero (see the seminal work [Alvarez et al.] and references therein for related discussions and links with DIN). In the paper, we consider equation (5), where the gradient  is replaced by the RMSProp direction. The purpose of this equation, which discretizes into INNAprop, was to preserve the “Newtonian structure” — the exponential steering idea — through its connection to the “limit equation”
$$a \tfrac{d}{dt} \mathrm{RMSProp}\, J(\theta) + b\, \mathrm{RMSProp}\, J(\theta) = 0.$$
This equation suggests indeed that Newtonian effects persist in the RMSProp-based setting, which motivates our formulation and the derivation from second-order systems (5). We will make this point explicit in the Appendix.

- **Experiments.** We conducted two supporting experiments (see supplement pdf attached):
  1. Using a SOTA framework for Muon, INNAProp ranks first, Muon second, and AdamW third (without any hyperparameter tuning of INNAProp).  Note that, Muon was released after the first version of the present work.
  2. An experiment showing that \(\alpha\) and the weight decay constant have distinct effects.

These elements will be included in a revised version of the paper.

---

### Decision · Action_Editor_tMhY · 2025-10-10

**Recommendation:** Accept with minor revision

**Additional Comments:**

In the rebuttal phase, the authors stated that they would clarify several points, in particular rewrite the derivation of the INNA algorithm. The two additional experiments performed in this time period (comparison with Muon and separataing the influences of $\alpha$ and the weight decay constant should also be included.

**Audience:**

Yes

**Audience Explanation:**

Given the considerable costs of training neural networks, there is an extremely wide body of research concerned with developing optimization algorithms for deep learning.
In this endeavor, handling second order information without resorting to prohibitive Hessian computation is a very interesting approach.
Finally, the code is released publicly, which is an essential step for reproducibility in numerical optimization for Machine Learning.

**Claims And Evidence:**

Yes

**Claims Explanation:**

The paper proposes INNAprop, a principled optimizer that appears to perform relatively well compared to AdamW on a variety of tasks and architecture.
INNAprop builds upon the Dynamic Inertial Newton (DIN) method, replacing the gradient in it by an RMSprop-based estimation.
A continuous time analysis is proposed, explaining why, though being 2nd order based, the algorithm does not resort to Hessian computation.

The exposition and methodology are generally deemed clear and transparent by reviewers.
Intuitive explanations and connections with existing works have been added in the rebuttal.

A point of attention is that Reviewer `MYuc` would like the paper to display SOTA performance, but:
- this is not a requirement of  TMLR, which prioritizes technical correction and scientific honesty over performance. In the paper's setting, expecting every optimization paper to display SOTA performance would have negative cherry-picking consequences.
- reviewers `Coz7` and `3aip` are positive about the paper's approach and clear exposition of strength and weaknesses